# MONICA: BENCHMARKING ON LONG-TAILED MEDICAL IMAGE CLASSIFICATION

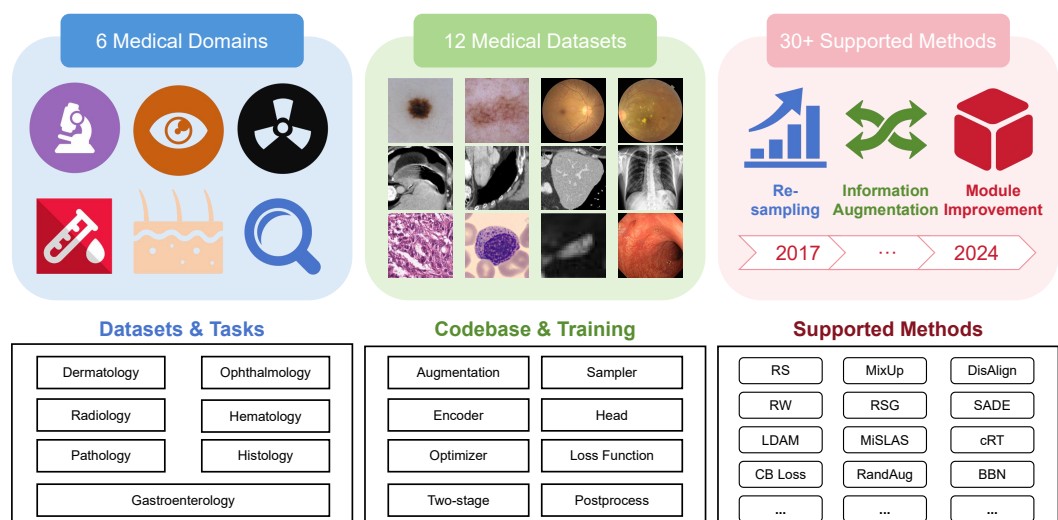

Figure 1: Overview of MONICA. The benchmark is meticulously designed for the evaluation of the generalization of various long-tailed learning methodologies on medical image classification. We also develop a unified, well-structured codebase integrating over **30** methods developed in relevant fields and evaluate which on **12** long-tailed medical datasets covering **6** medical domains.

## ABSTRACT

Long-tailed learning is considered to be an extremely challenging problem in data imbalance learning. It aims to train well-generalized models from a large number of images that follow a long-tailed class distribution. In the medical field, many diagnostic imaging exams such as dermoscopy and chest radiography yield a long-tailed distribution of complex clinical findings. Recently, long-tailed learning in medical image analysis has garnered significant attention. However, the field currently lacks a unified, strictly formulated, and comprehensive benchmark, which often leads to unfair comparisons and inconclusive results. To help the community improve the evaluation and advance, we build a unified, well-structured codebase called **M**edical **O**pe**N**-source Long-ta**I**led Classifi**CA**tion (**MONICA**), which implements over **30** methods developed in relevant fields and evaluated on **12** long-tailed medical datasets covering **6** medical domains. Our work provides valuable practical guidance and insights for the field, offering detailed analysis and discussion on the effectiveness of individual components within the inbuilt state-of-the-art methodologies. We hope this codebase serves as a comprehensive and reproducible benchmark, encouraging further advancements in long-tailed medical image learning. The codebase will be publicly available on GitHub.

## 1 INTRODUCTION

The deep learning techniques have proven effective for most computer vision tasks benefiting from the grown-up dataset scale (Deng et al., 2009; Dosovitskiy et al., 2020; He et al., 2016). However,

training a well-generalized deep-learning-based model is data-driven and requires a balanced data distribution. In the real world, the collected image datasets often exhibit a long-tailed distribution due to the complex findings and attributes (Ridnik et al., 2021; Lin et al., 2014). Models trained from such datasets always result in the overprediction of head classes and underprediction of tail classes (Liu et al., 2019; Zhang et al., 2023a). In the medical domain, medical image datasets are typically long-tailed because of the natural frequency of diseases in the population and the challenges in collecting sufficient samples of rare conditions (Ju et al., 2023). However, it is vital to recognize these rare diseases in real-world practice, as they are relatively rare for doctors and may also lack diagnostic capacity.

To address the long-tailed class imbalance, massive deep long-tailed learning studies have been conducted for natural image recognition. A recent survey (Zhang et al., 2023a) grouped existing methods into three main categories based on their main technical contributions, i.e., re-sampling (Chawla et al., 2002; Estabrooks et al., 2004; Liu et al., 2008; Zhang & Pfister, 2021), information augmentation (Zhang, 2017; Zhong et al., 2021; Li et al., 2022) and module improvement (Kang et al., 2020; Tang et al., 2020; Zhang et al., 2021). Class Re-sampling aims to balance the distribution by over-sampling the minority-class samples or under-sampling the majority-class samples. Data Augmentation aims to enhance the size and quality of datasets by applying predefined transformations to each data/feature for model training. Module improvement aims to modify the network to better learn from a long-tailed distribution with a specific module design. In Sec. 3.2, we will briefly introduce these methodologies supported in our codebase as groups.

In recent years, research on long-tailed medical image classification (LTMIC) has garnered significant attention. Current research on LTMIC is primarily focused on dermatology (Ju et al., 2022; Roy et al., 2022; Mehta et al., 2022; Zhang et al., 2023b), ophthalmology (Ju et al., 2021; 2023; Li et al., 2024), and radiology (Holste et al., 2022; 2024; Jeong et al., 2023), where more abundant datasets and well-defined diagnostic tasks are available. However, these methods are often evaluated under varying experimental settings, making it difficult to comprehensively compare and select the best approach for practical applications. This challenge is further compounded by the lack of standard benchmarks and pipelines. Overall, these works may share common shortcomings, which arise from the following factors. **1) Datasets.** Existing works on LTMIC are evaluated on different datasets. Despite the specialized nature of medical data, we are still curious to explore whether there are highly generalizable methods that can perform well across different datasets, tasks, or medical domains. **2) Partition Schemes.** The partition schemes are vita important for long-tailed learning to ensure a fair comparison and metric evaluation. For example, in natural image long-tailed learning, although the training data follows a long-tailed distribution, the test set is often balanced. However, in medical imaging, the test set typically mirrors the distribution of the training set. As a result, relying solely on overall accuracy may not accurately reflect the performance of trained models. **3) Comparison Methodologies** The methodologies used for comparing different approaches in LTMIC can vary significantly, which adds another layer of complexity when trying to assess their effectiveness. Due to the lack of standardized comparison practices, such as consistent use of baseline models, evaluation metrics, and reporting standards, it becomes challenging to draw meaningful conclusions across studies. Furthermore, the varying availability of codes and experiment replication further hinder the transparency of fair comparison and the ability to establish a clear understanding of which methods are truly superior. These inconsistencies highlight the need for more unified and comprehensive comparison methodologies in future research.

Our main contributions are summarized as follows: (1) We introduce MONICA, the first comprehensive LTMIC benchmark, where **30+** methodologies from relevant fields are impartially evaluated from scratch across **12** long-tailed medical datasets spanning **6** medical domains. This benchmark covers datasets with varying scales, granularity, and imbalance ratios, offering a robust framework for researchers to objectively assess their models against a wide array of baselines, providing a clear measurement of each method's effectiveness. (2) We developed a well-structured codebase specifically for customized LTMIC. The framework is modular, featuring decoupled components such as augmentations, well-known backbones, loss functions, optimization strategies, and distributed training. This codebase offers best practices for researchers and engineers to identify applicable methodologies for both pre-defined benchmarks and their own customized datasets. (3) We performed extensive empirical analyses and gave a detailed discussion on valuable insights that suggest promising directions for methodological and evaluation innovations in future LTMIC research.

## 2 SUPPORTED TASKS, BENCHMARKS, AND METRICS

### 2.1 PROBLEM DEFINITION AND SUPPORTED TASKS

LTMIC seeks to learn a deep neural network model from a training dataset with a long-tailed class distribution. Let $\{x_i, y_i\}_{i=1}^n$ be the long-tailed training set, where each sample $x_i$ has a corresponding class label $y_i$. The total number of training set over $K$ classes is $n = \Sigma_{k=1}^K n_k$, where $n_k$ denotes the data number of class $k$; let $\pi$ denote the vector of label frequencies, where $\pi = \frac{n_k}{n}$ indicates the label frequency of class $k$ where $\rho$ denoted as imbalance ratio $\rho = \frac{n_1}{n_k}$. Without loss of generality, a common assumption in long-tailed learning is when the classes are sorted by cardinality in decreasing order (i.e., if $i_1 < i_2$, then $n_{i_1} > n_{i_2}$, and $n_1 \gg n_k$ ). For a multi-label setting, the class label $y_i$ would be a set of Bernoulli distribution $y_i \in \{0,1\}^k$. Label cardinality $L_{Card}(S) = \frac{1}{n}\Sigma_{i=1}^n |y_i|$ is commonly used to describe the degree of label co-occurrence in a multi-label dataset. In this context, MONICA is developed to support the training and evaluation of both single-label / multi-class (MC) and multi-label (ML) long-tailed learning on conducted benchmarks and customized datasets.

### 2.2 BENCHMARKS AND DATASETS

Table 1: Comparison, partition, and statistics of datasets across various medical specialties.

| Dataset | Dermatology | | Ophthalmology | | Radiology | | Pathology | Hematology | Histology | Gastroenterology |
|---|---|---|---|---|---|---|---|---|---|---|
| | ISIC-2019-LT | DermaMNIST | ODIR | RFMiD | OrganA/C/SMNIST | CheXpert | PathMNIST | BloodMNIST | TissueMNIST | KVASIR |
| Data Modality | Dermatoscope | Dermatoscope | Fundus | Fundus | CT | X-Ray | Pathology | Microscope | Microscope | Endoscope |
| Task | MC | MC | ML | ML | MC | ML | MC | MC | MC | MC |
| Class Number | 8 | 7 | 12 | 29 | 11 | 14 | 9 | 8 | 8 | 14 |
| Imbalance Ratio | 100 / 200 / 500 | 100 | 80 | 310 | 100 | 33 | 100 | 100 | 100 | 20 |
| Train Samples | 10,322 / 9,400 / 8,494 | 6,964 | 7,000 | 1,920 | 16,597 / 7,712 / 9,146 | 178,731 | 29,276 | 4,809 | 109,532 | 4,656 |
| Validation Samples | 400 | 1003 | 1,000 | 640 | 6,491 / 2,392 / 2,452 | 44,683 | 10,004 | 1,712 | 23,640 | 700 |
| Test Samples | 800 | 2005 | 2,000 | 640 | 17,778 / 8,216 / 8,827 | 243 | 7,180 | 3,421 | 47,280 | 1400 |
| Group Split | 2 / 5 / 8 | 1 / 5 / 7 | 3 / 9 / 12 | 6 / 14 / 29 | 3 / 6 / 11 | 4 / 10 / 14 | 2 / 5 / 9 | 3 / 5 / 8 | 3 / 5 / 8 | 4 / 8 / 14 |

To address the challenge of long-tailed medical image classification, we conducted our benchmark on 12 datasets covering Dermatology (ISIC-2019 (Tschandl et al., 2018), Dermamnist (Yang et al., 2023)), Ophthalmology (ODIR (ODIR), RFMiD (Quellec et al.)), Radiology (Organamnist, Organamnist, Organamnist (Yang et al., 2023), CheXpert (Irvin et al., 2019)), Pathology (Pathmnist), Hematology (Bloodmnist), Histology (Tissuemnist) (Yang et al., 2023) and Gastroenterology (KVASIR) (Pogorelov et al., 2017). To evaluate the ability of existing methodologies under extremely challenging imbalance conditions, some widely-used medical image datasets and tasks with fewer than 8 categories such as Diabetic Retinopathy (Li et al., 2019) Grading are not considered.

#### 2.2.1 DERMATOLOGY DATASETS

**ISIC-2019-LT** (Ju et al., 2022) is a long-tailed version constructed from ISIC-2019 Challenge (Tschandl et al., 2018), which aims to classify 8 kinds of diagnostic categories. We follow FlexSampling (Ju et al., 2022) and sample a subset from a Pareto distribution. With $k$ classes and imbalance ratio $r = \frac{N_0}{N_{k-1}}$, the number of samples for class $c \in [0, k)$ can be calculated as $N_c = \left(r^{-(k-1)}\right)^c * N_0$. We set $r = \{100, 200, 500\}$ for three different imbalance levels. We select 50 and 100 images from the remained samples as validation set and test set.

**DermaMNIST** is created by MedMNIST (Yang et al., 2023) based on the HAM10000 (Tschandl et al., 2018), a large collection of multi-source dermatoscopic images of common pigmented skin lesions. The dataset consists of 10, 015 dermatoscopic images categorized as 7 different diseases, formalized as a multi-class classification task. The original images are split into training, validation and test set with a ratio of 7: 1: 2. We modify the training set with Pareto distribution and imbalance ratio of $r = 100$. We keep the use of original validation and test set for evaluation.

#### 2.2.2 OPHTHALMOLOGY DATASETS

**Ocular Disease Intelligent Recognition (ODIR)** (ODIR) is a structured ophthalmic database of 5,000 patients with age, colour fundus photographs of left and right eyes, and doctors' diagnostic keywords. Specifically, the ODIR dataset was originally divided into 8,000 / 1,000 / 2,000 images for training/off-site testing / on-site testing. The classes of annotations can be divided into two levels:

coarse and fine. There are 8 classes at the coarse level, where one or more conditions are given on a patient-level diagnosis, resulting in a multi-label classification challenge.

**RFMiD** dataset (Quellec et al.) consists of 3200 fundus images captured using three different fundus cameras with 46 conditions annotated through adjudicated consensus of two senior retinal experts. The RFMiD dataset was originally divided into 1,920 / 640 / 640 images for training/validation/testing. We followed the setting in the RFMiD challenge, the diseases with more than 10 images belong to an independent class and all other disease categories are merged as "OTHER". This finally constitutes 29 classes (normal + 28 diseases or lesions) for disease classification.

### 2.2.3 RADIOLOGY DATASETS

**OrganA, C, SMNIST** is based on 3D computed tomography (CT) images from Liver Tumor Segmentation Benchmark (LiTS) (Bilic et al., 2023). They are renamed from OrganMNIST-Axial/Coronal/Sagittal (in MedMNIST (Yang et al., 2023)), which uses bounding-box annotations of 11 body organs from another study to obtain the organ labels. 2D images are cropped from the center slices of the 3D bounding boxes in axial/coronal/sagittal views. All three datasets contain the labeling of 11 body organs, resulting in a multi-class classification task. We modify the training set with Pareto distribution and imbalance ratio of $r = 100$. We keep the use of original validation and test set for evaluation.

**CheXpert** dataset (Irvin et al., 2019) is a large dataset that contains 224,316 chest radiographs with 14 kinds of observations. The training labels in the dataset for each observation are either 0 (negative), 1 (positive), or u (uncertain). For convenience, we map all uncertain instances to 0 (negative). The original images are split into training, validation and test set with a ratio of 7: 1: 2.

### 2.2.4 OTHERS

**PathMNIST** (Yang et al., 2023) is constructed for predicting survival from colorectal cancer histology slides, providing a dataset (NCT-CRC-HE-100K) (Kather et al., 2018) of 100, 000 non-overlapping image patches from hematoxylin & eosin stained histological images, and a test dataset (CRC-VAL-HE-7K) of 7, 180 image patches from a different clinical center. The dataset is comprised of 9 types of tissues as a multi-class classification task. We modify the training set with Pareto distribution and imbalance ratio of $r = 100$. We keep the use of original validation and test set for evaluation.

**BloodMNIST** (Yang et al., 2023) is based on a dataset of individual normal cells, captured from individuals without infection, hematologic or oncologic disease and free of any pharmacologic treatment at the moment of blood collection. It contains a total of 17,092 images and is organized into 8 classes. The source dataset was originally split to a ratio of 7 : 1 : 2 into training, validation and test set. We modify the training set with Pareto distribution and imbalance ratio of $r = 100$. We keep the use of original validation and test set for evaluation.

**TissueMNIST** (Yang et al., 2023) is based on the Broad Bioimage Benchmark Collection (Ljosa et al., 2012). The dataset contains 236, 386 human kidney cortex cells, segmented from 3 reference tissue specimens and organized into 8 categories. The source dataset was originally split to a ratio of 7: 1: 2 into training, validation and test set. We modify the training set with Pareto distribution and imbalance ratio of $r = 100$. We keep the use of original validation and test set for evaluation.

**Kvasir** (Pogorelov et al., 2017) is a long-tailed dataset of 10,662 gastrointestinal tract images with 23 classes from different anatomical and pathological landmarks. We modify the original dataset with Pareto distribution and imbalance ratio of $r = 20$. Those categories with images less than 50 are not included. We select 50 and 100 images from the remained samples as validation set and test set.

## 3 MONICA AND SUPPORTED METHODOLOGIES

### 3.1 CODEBASE STRUCTURE

The whole training process here is fragmented into multiple components, including augmentation (.MONICA.dataset), sampling strategies (.MONICA.sampler), model architectures (.MONICA.models), and loss functions (.MONICA.losses) etc. For instance, vision models are decoupled into several encoders and classification heads according to different methodology designs. This

modular architecture allows researchers to easily craft different counterparts as customized datasets and tasks are needed. With the help of configuration files in .MONICA.configs, users can tailor specialized visual classification models and their associated training strategies with ease.

## 3.2 SUPPORTED METHODOLOGIES

According to a recent survey (Zhang et al., 2023a), we group existing methods into three main categories based on their main technical contributions, i.e., class re-sampling , information augmentation and module improvement . Based on factors such as the availability of open-source code, its impact, and ease of implementation, we have selected and introduced over 30 methodologies supported in MONICA. Note that one methodology may contain more than one of those three taxonomy and we will group them based on their primary motivation and technical contribution when presenting them.

### 3.2.1 RE-SAMPLING METHODOLOGIES

**Re-sampling** aims to balance the distribution by over-sampling the minority-class samples or under-sampling the majority-class samples following designed schemes. **Focal loss** (Lin et al., 2017) is designed to down-weight the loss assigned to well-classified examples, focusing more on hard-to-classify instances. **Class-balanced (CB) loss** (Cui et al., 2019) addresses class imbalance by weighting the loss inversely proportional to the effective number of samples per class, thereby reducing the impact of over-represented classes. **LADE loss** (Hong et al., 2021) proposed to use them to post-adjust model outputs so that the trained model can be calibrated for arbitrary test class distributions. **LDAM loss** (Cao et al., 2019) adjusts the margins for different classes based on their label distribution, promoting larger margins for underrepresented classes to improve their classification accuracy. **EQL** (Tan et al., 2020) directly down-weights the loss values of tail-class samples when they serve as negative labels for head-class samples. **Balanced softmax** (Jiawei et al., 2020) proposed to adjust prediction logits by multiplying by the label frequencies, so that the bias of class imbalance can be alleviated by the label prior before computing final losses. **VS loss** (Kini et al., 2021) intuitively analyzed the distinct effects of additive and multiplicative logit-adjusted losses, leading to a novel VS loss to combine the advantages of both forms of adjustment.

### 3.2.2 INFORMATION AUGMENTATION METHODOLOGIES

Data Augmentation aims to enhance the size and quality of datasets by applying predefined transformations to each data/feature for model training. **MixUp** (Zhang, 2017) is proposed to improve the model generalization but found to be effective for long-tailed learning by information shared between head and tailed classes. **MiSLAS** (Zhong et al., 2021) proposed to enhance the representation learning with data mixup in the first stage, while applying a label-aware smoothing strategy for better classifier generalization in the second stage. **RSG** (Wang et al., 2021a) proposed to dynamically estimate a set of feature centers for each class, and use the feature displacement between head-class sample features and their nearest intra-class feature center to augment each tail sample feature. **RIDE** (Wang et al., 2021b) introduced a knowledge distillation method to reduce the parameters of the multi-expert model by learning a student network with fewer experts. **GCL loss** (Li et al., 2022) perturbs logits with Gaussian noise of varying amplitudes, especially larger for tail classes. Additionally, a class-based effective number sampling strategy with classifier re-training is proposed to mitigate classifier bias.

### 3.2.3 MODULE IMPROVEMENT METHODOLOGIES

Decoupling (Kang et al., 2020; Zhou et al., 2020) was the pioneering work to introduce such a two-stage decoupled training scheme. It empirically evaluated different sampling strategies for representation learning in the first stage and then evaluated different classifier training schemes by fixing the trained feature extractor in the second stage. In the classifier learning stage, there are also four methods, including **classifier re-training** with class-balanced sampling, the **nearest class mean classifier**, the $\tau$**-normalized classifier**, and the **learnable weight-scaling classifier**. **Range loss** (Zhang et al., 2017) is designed to reduce overall intrapersonal variations while enlarging interpersonal differences simultaneously. **Causal classifier** (Tang et al., 2020) resorted to causal inference for keeping the good and removing the bad momentum causal effects in long-tailed learning.

Table 2: The comparison study results on ISIC-2019-LT benchmark.

| | ISIC-2019-LT | | | | | | | | | | | |
|---|---|---|---|---|---|---|---|---|---|---|---|---|
| Imbalance Ratio | $r = 100$ | | | | $r = 200$ | | | | $r = 500$ | | | |
| Methods | Head | Medium | Tail | Avg. | Head | Medium | Tail | Avg. | Head | Medium | Tail | Avg. |
| ERM | 79.00 | 60.67 | 38.33 | 59.33 | 78.50 | 56.67 | 27.00 | 54.06 | 78.00 | 46.67 | 12.67 | 45.78 |
| RS | 69.50 | 61.33 | 49.33 | 60.06 | 76.00 | **62.67** | 36.33 | 58.33 | 78.50 | 44.00 | 19.67 | 47.39 |
| RW | 68.00 | 55.33 | 53.67 | 59.00 | 73.50 | 54.67 | 41.33 | 56.50 | 62.00 | 38.00 | 34.33 | 44.78 |
| Focal | 73.50 | 54.00 | 44.33 | 57.28 | 79.50 | 53.00 | 31.00 | 54.50 | **83.00** | 44.00 | 13.00 | 46.67 |
| CB-Focal | 71.00 | 57.67 | 52.67 | 60.44 | 72.50 | 52.00 | 51.00 | 58.50 | 59.50 | 42.33 | 43.33 | 48.39 |
| LADELoss | 78.50 | 52.33 | 43.67 | 58.17 | **84.00** | 52.33 | 18.67 | 51.67 | 78.50 | 43.00 | 14.00 | 45.17 |
| LDAM | 78.50 | 55.67 | 41.67 | 58.61 | 81.50 | 52.33 | 31.00 | 54.94 | 76.50 | 41.33 | 19.33 | 45.72 |
| BalancedSoftmax | 62.50 | 54.33 | 61.00 | 59.28 | 77.00 | 53.67 | 55.00 | 61.89 | 62.00 | 49.67 | 41.67 | 51.11 |
| VSLoss | 80.00 | 56.33 | 33.67 | 56.67 | 80.50 | 51.00 | 28.67 | 53.39 | 79.00 | 47.67 | 11.00 | 45.89 |
| MixUp | 78.00 | 50.67 | 35.67 | 54.78 | 83.00 | 46.67 | 21.33 | 50.33 | 76.00 | 48.00 | 9.00 | 44.33 |
| MiSLAS | 57.50 | 52.33 | 57.67 | 55.83 | 71.50 | 48.33 | 49.67 | 56.50 | 63.50 | 43.00 | 39.33 | 48.61 |
| GCL | 57.50 | **63.33** | **71.33** | 64.06 | 71.00 | 56.67 | **64.33** | **64.00** | 63.50 | **55.00** | **46.00** | **54.83** |
| cRT | 74.50 | 59.67 | 55.67 | 63.28 | 81.00 | 60.00 | 39.67 | 60.22 | 63.50 | 48.33 | 28.00 | 46.61 |
| LWS | 72.50 | 52.67 | 45.33 | 56.83 | 79.50 | 51.33 | 32.67 | 54.50 | 68.00 | 43.33 | 29.67 | 47.00 |
| KNN | 70.00 | 55.67 | 58.67 | 61.44 | 77.00 | 51.33 | 46.67 | 58.33 | 75.00 | 45.33 | 27.33 | 49.22 |
| LAP | 75.00 | 59.33 | 49.33 | 61.22 | 76.50 | 50.67 | 48.00 | 58.39 | 72.00 | 43.67 | 33.00 | 49.56 |
| De-Confound | 79.00 | 52.33 | 47.67 | 59.67 | 82.50 | 52.67 | 24.33 | 53.17 | 72.50 | 45.33 | 9.67 | 42.50 |
| DisAlign | 81.00 | 60.00 | 52.33 | **64.44** | 78.00 | 59.67 | 52.33 | 63.33 | 68.50 | 49.33 | 37.33 | 51.72 |
| BBN | **82.50** | 58.33 | 46.00 | 62.28 | 76.50 | 62.33 | 31.00 | 54.94 | 75.50 | 50.67 | 19.00 | 48.39 |

**DisAlign** (Zhang et al., 2021) innovated the classifier training with a new adaptive logits adjustment strategy. **BBN** (Zhou et al., 2020) proposed to use two network branches, i.e., a conventional learning branch and a re-balancing branch, to handle long-tailed recognition. **SADE** (Li et al., 2021) explored a new multi-expert scheme to handle test-agnostic long-tailed recognition, where the test class distribution can be either uniform, long-tailed or even inversely long-tailed. **SAM** (Foret et al., 2020) minimizes both loss value and loss sharpness by seeking parameters in neighborhoods.

# 4 EXPERIMENTS

## 4.1 IMPLEMENTATION DETAILS

We implement all experiments in PyTorch, ensuring a fair comparison by using unified settings with consistent hyperparameters and architecture choices, unless otherwise specified in the paper. For instance, we use ResNet-50 (He et al., 2016) as the primary network backbone across all methods, modifying it as needed for certain module improvement methods like BBN. Model training is conducted with the Adam optimizer, using a batch size of 256, a learning rate of 0.0003, and an input size of 224×224, except for CheXpert, where the input size is 512×512. For certain methodologies, such as SAM, we adhere to the specific optimizer and hyperparameters following the original paper. All these designs are for the fairness and the practicality of the comparison on the benchmark. We use top-1 accuracy to evaluate the performance of single-label datasets and mean average precision is adopted for multi-label datasets following DBLoss (Wu et al., 2020). The main benchmark development and testing are performed using $8 \times$ NVIDIA RTX4090 GPUs.

## 4.2 MAIN RESULTS

We conducted extensive experiments on various datasets in the comparison of the state-of-the-art long-tailed learning methods. We introduced and implemented 30+ methods but only present results for the most relevant ones to avoid redundancy and maintain clarity. Methods with similar or suboptimal performance were excluded to focus on those that best support our key findings within the main motivation's constraints.

**Overall performance evaluation.** Table 2 and Table 3 compare the performance of various methods on the ISIC-2019-LT, MedMNIST, and KVASIR benchmarks under different imbalance ratios and datasets. We group the methods as introduced in Sec. 3.2 and filter out these results which do not outperform ERM in terms of any metrics. ERM, as a baseline, generally struggles with tail classes, showing declining performance as imbalance increases. Re-sampling (RS) and re-weighting (RW) methods improve tail class performance but still face challenges under extreme imbalance. Advanced methods like GCL and MiSLAS demonstrate strong results, particularly in handling severe class

Table 3: The comparison study results on MedMNIST and KVASIR benchmarks.

| | BloodMNIST | | | | DermaMNIST | | | | PathMNIST | | | | TissueMNIST | | | |
|---|---|---|---|---|---|---|---|---|---|---|---|---|---|---|---|---|
| Methods | Head | Medium | Tail | Avg. | Head | Medium | Tail | Avg. | Head | Medium | Tail | Avg. | Head | Medium | Tail | Avg. |
| ERM | 97.27 | 97.16 | 82.00 | 92.14 | 95.82 | 61.86 | 44.08 | 67.25 | 98.46 | 99.64 | 83.07 | 93.72 | 64.38 | 62.11 | 23.21 | 49.90 |
| RS | 95.75 | 99.20 | 91.17 | 95.37 | 91.95 | 65.75 | 59.75 | 72.48 | 99.26 | 97.47 | 86.78 | 94.50 | 50.26 | 64.71 | 52.88 | 55.95 |
| RW | 97.39 | 99.04 | 88.14 | 94.86 | 87.32 | 65.31 | 69.19 | 73.94 | 96.62 | 99.74 | 88.57 | 94.97 | 59.72 | 66.15 | 43.44 | 56.44 |
| Focal | 97.09 | 99.42 | 80.75 | 92.42 | 95.08 | 59.60 | 60.57 | 71.75 | 98.66 | 98.66 | 87.51 | 94.18 | 61.89 | 63.17 | 22.07 | 49.04 |
| CB-Focal | 96.97 | 98.34 | 85.21 | 93.51 | 84.41 | 63.27 | 73.99 | 73.89 | 98.44 | 99.45 | 84.10 | 94.00 | 58.60 | 67.08 | 45.31 | 56.99 |
| LADELoss | 97.16 | 98.77 | 72.76 | 89.56 | 91.28 | 67.43 | 50.60 | 69.77 | 98.55 | 98.48 | 84.43 | 93.82 | 65.96 | 63.33 | 23.92 | 51.07 |
| LDAM | 96.74 | 98.45 | 87.32 | 94.17 | 92.62 | 63.71 | 53.15 | 69.82 | 97.37 | 98.66 | 86.07 | 94.03 | 63.31 | 64.57 | 14.64 | 47.51 |
| BalancedSoftmax | 96.55 | 98.13 | 90.71 | 95.13 | 89.41 | 67.35 | 76.61 | 77.79 | 96.39 | 99.54 | 88.42 | 94.78 | 53.86 | 68.41 | 52.91 | 58.39 |
| VSLoss | 96.05 | 97.70 | 86.59 | 93.45 | 92.84 | 64.75 | 49.70 | 69.10 | 98.87 | 98.31 | 83.46 | 93.54 | 66.78 | 61.45 | 21.54 | 49.92 |
| MixUp | 98.04 | 98.93 | 79.93 | 92.30 | 95.90 | 59.03 | 54.43 | 69.78 | 99.38 | 99.54 | 84.72 | 94.54 | 65.85 | 62.79 | 13.54 | 47.39 |
| MiSLAS | 93.15 | 99.25 | 81.74 | 91.38 | 65.92 | 67.53 | 75.72 | 69.72 | 96.30 | 98.99 | 88.63 | 94.64 | 42.84 | 64.71 | 49.91 | 52.49 |
| GCL | 96.98 | 99.52 | 90.70 | 95.74 | 68.75 | 73.11 | 90.03 | 77.30 | 96.16 | 99.89 | 87.80 | 94.62 | 64.44 | 69.99 | 33.69 | 56.04 |
| cRT | 97.64 | 99.36 | 87.43 | 94.81 | 88.52 | 73.36 | 70.09 | 77.32 | 93.86 | 99.69 | 90.59 | 94.71 | 59.47 | 67.83 | 38.70 | 55.33 |
| LWS | 90.30 | 75.36 | 1.50 | 55.72 | 44.87 | 7.82 | 1.26 | 17.98 | 59.44 | 32.10 | 23.94 | 38.49 | 59.28 | 33.59 | 0.60 | 31.16 |
| KNN | 93.63 | 95.88 | 83.31 | 90.94 | 85.31 | 66.62 | 59.67 | 70.53 | 93.56 | 99.79 | 89.67 | 94.34 | 54.77 | 55.40 | 51.38 | 53.85 |
| De-Confound | 97.14 | 97.49 | 85.10 | 93.24 | 95.08 | 61.54 | 49.25 | 68.62 | 93.45 | 99.85 | 88.53 | 93.94 | 65.32 | 61.35 | 24.34 | 50.34 |
| DisAlign | 97.11 | 98.98 | 88.82 | 94.97 | 85.68 | 66.04 | 75.72 | 75.81 | 96.70 | 97.95 | 91.85 | 95.50 | 63.20 | 59.73 | 60.21 | 56.71 |
| BBN | 96.24 | 98.23 | 89.39 | 94.62 | 91.87 | 70.98 | 64.02 | 75.62 | 98.14 | 98.95 | 90.33 | 95.80 | 58.62 | 69.68 | 43.29 | 57.19 |

| | OrganAMNIST | | | | OrganCMNIST | | | | OrganSMNIST | | | | KVASIR | | | |
|---|---|---|---|---|---|---|---|---|---|---|---|---|---|---|---|---|
| Methods | Head | Medium | Tail | Avg. | Head | Medium | Tail | Avg. | Head | Medium | Tail | avg | Head | Medium | Tail | Avg. |
| ERM | 82.67 | 78.20 | 68.56 | 76.48 | 93.50 | 59.20 | 65.18 | 72.63 | 88.33 | 56.89 | 66.90 | 70.71 | 95.50 | 93.25 | 60.33 | 83.03 |
| RS | 76.14 | 82.10 | 63.39 | 73.88 | 92.49 | 57.14 | 64.06 | 71.23 | 89.99 | 52.61 | 66.39 | 69.66 | 95.75 | 94.50 | 62.83 | 84.36 |
| RW | 83.79 | 78.89 | 73.55 | 78.74 | 92.20 | 68.91 | 66.84 | 75.98 | 83.01 | 58.68 | 69.60 | 70.43 | 96.75 | 88.75 | 67.67 | 84.39 |
| Focal | 85.08 | 73.19 | 68.88 | 75.72 | 91.98 | 57.02 | 63.36 | 70.79 | 86.24 | 54.93 | 64.77 | 68.65 | 95.75 | 92.00 | 57.67 | 81.81 |
| CB-Focal | 75.95 | 80.52 | 74.12 | 76.86 | 94.16 | 57.99 | 67.36 | 73.17 | 84.18 | 60.42 | 71.42 | 72.00 | 95.00 | 84.75 | 71.50 | 83.75 |
| LADELoss | 84.54 | 77.00 | 68.73 | 76.76 | 91.22 | 61.58 | 62.86 | 71.88 | 83.32 | 59.04 | 64.92 | 69.09 | 96.25 | 90.00 | 60.83 | 82.36 |
| LDAM | 82.86 | 76.38 | 70.14 | 76.46 | 90.98 | 61.68 | 69.05 | 73.94 | 86.07 | 56.60 | 71.11 | 71.90 | 96.00 | 89.00 | 63.17 | 82.72 |
| BalancedSoftmax | 79.45 | 76.48 | 72.79 | 76.24 | 93.58 | 62.68 | 67.61 | 74.62 | 81.44 | 59.27 | 70.05 | 70.26 | 95.25 | 88.25 | 65.83 | 83.11 |
| VSLoss | 81.59 | 81.73 | 67.85 | 77.06 | 94.79 | 61.42 | 67.08 | 74.43 | 86.46 | 58.66 | 66.70 | 70.61 | 96.75 | 92.75 | 62.83 | 84.11 |
| MixUp | 85.54 | 77.78 | 65.60 | 75.97 | 92.25 | 57.43 | 66.97 | 72.22 | 89.15 | 50.09 | 64.25 | 67.83 | 96.25 | 92.25 | 55.33 | 81.28 |
| MiSLAS | 73.78 | 73.71 | 67.63 | 71.71 | 84.86 | 51.08 | 64.40 | 66.78 | 74.66 | 52.74 | 69.39 | 65.60 | 94.25 | 85.75 | 71.00 | 83.67 |
| GCL | 75.72 | 83.69 | 77.86 | 79.09 | 94.15 | 60.46 | 68.99 | 74.54 | 88.64 | 62.68 | 73.08 | 74.80 | 96.00 | 93.75 | 65.67 | 85.14 |
| cRT | 83.88 | 76.10 | 67.34 | 75.77 | 92.11 | 63.17 | 67.38 | 74.22 | 88.23 | 58.99 | 69.79 | 72.34 | 95.50 | 88.00 | 68.83 | 84.11 |
| LWS | 84.37 | 35.86 | 7.49 | 42.57 | 44.87 | 7.82 | 1.26 | 17.98 | 59.44 | 32.10 | 23.94 | 38.49 | 95.00 | 87.00 | 61.67 | 81.22 |
| KNN | 79.03 | 78.74 | 68.64 | 75.47 | 88.66 | 56.54 | 65.97 | 70.39 | 85.33 | 54.82 | 66.60 | 68.92 | 95.00 | 91.50 | 67.00 | 84.50 |
| De-Confound | 77.50 | 87.72 | 68.87 | 78.03 | 92.56 | 64.81 | 64.58 | 73.98 | 87.03 | 61.35 | 66.07 | 71.48 | 96.25 | 90.50 | 66.50 | 84.42 |
| DisAlign | 80.51 | 78.31 | 69.20 | 76.01 | 93.74 | 62.69 | 72.65 | 76.36 | 88.09 | 58.77 | 66.50 | 71.12 | 95.75 | 89.25 | 68.50 | 84.50 |
| BBN | 74.63 | 77.48 | 67.08 | 73.06 | 91.55 | 63.41 | 63.86 | 72.94 | 87.91 | 59.28 | 71.74 | 72.98 | 95.25 | 89.25 | 70.00 | 84.83 |

imbalance, with GCL consistently showing the highest average performance across multiple datasets. Methods like LADELoss, LDAM, and PriorCELoss are effective in improving tail performance, while VSLoss and BalancedSoftmax also perform well, particularly in medium and head classes. In the following sections, we will explore the factors that contribute to the effectiveness of these methods and offer insights into best practices.

**Curse of shot-based group evaluation.** Improving the performance on tail and medium groups often comes at the cost of reducing accuracy on the head group. For instance, in the ISIC dataset with an imbalance ratio of 100, while the average performance remains similar for GCL and DisAlign, GCL sacrifices a substantial portion of the head group's performance (from 79.00% to 57.50%) to achieve a superior improvement in the tail group's performance (from 38.33% to 71.33%). While the trade-off is inevitable, it presents an additional challenge in designing novel metrics that can globally assess performance to meet the demands of real-world practice, particularly in complex disease systems.

**Effectiveness of re-sampling across SOTAs.** The primary challenge in long-tailed problems is underfitting tail classes due to data imbalance. While resampling is a simple and straightforward approach that shows promising improvements across various datasets and tasks. Also, it is easily incorporated in most state-of-the-art methods along with other module improvements. In this context, further improvements could involve replacing or combining re-sampling modules with alternative approaches, as MONICA offers decoupled training to facilitate these explorations. Given the diverse implementations and intersections of techniques (e.g., Focal Loss as a component in CBLoss), we summarize their key characteristics—class weighting, modulating factors, and loss formulas—in Table 4 to provide a comprehensive overview of how each method addresses class imbalance through class- or instance-level sampling strategies.

**MixUp can improve the feature representation** MixUp (Zhang, 2017), while often included in data augmentation strategies, does not always improve performance for long-tailed learning when used alone, as shown in the results. However, it shows value in blending head and tail data, facilitating knowledge transfer, reducing head class prediction confidence, and enhancing feature encoder generalization, as evidenced in two-stage decoupling work (Zhong et al., 2021; Li et al., 2022). MiSLAS (Zhong et al., 2021) also empirically observed that data mixup is beneficial to feature learning but has a negative effect on classifier training under the two-stage training scheme. Therefore, assessing MixUp based solely on performance is not fair; integrating it with other methods,

Table 4: Overview of various loss functions based on re-sampling strategies.

| Loss Function | Class Weighting | Modulating Factor | Formula |
|---|---|---|---|
| ERM (Original Sampling) | None (uses original data distribution) | None | $\text{CrossEntropy}(\text{logits}, y)$ |
| Re-balanced Sampling | $\frac{1}{n_c}$, where $n_c$ is the number of samples in class $c$ | None | $\text{CrossEntropy}(\text{logits}, y)$ with $P(y=c) \propto \frac{1}{n_c}$ |
| Difficulty-based Sampling | Learning difficulty, typically inverse of validation accuracy $a_c$ | None | $\text{CrossEntropy}(\text{logits}, y)$ with $P(y=c) \propto \frac{1}{a_c}$ |
| Focal Loss | Optional per-sample weighting via $\alpha$ | $(1-p_t)^\gamma$, where $p_t$ is the probability of correct classification | $-\alpha(1-p_t)^\gamma \log(p_t)$ |
| CBLoss | The number of samples per class $\frac{1-\beta}{1-\beta^{n_c}}$ | None | $\text{weight} \times \text{Loss}(p_t, y)$ |
| LADELoss | Class weighting based on the number of samples per class | Regularization using ReMINE to prevent overfitting | $-\sum(\text{ReMINE\_loss} \times \text{cls\_weight})$ |
| LDAM Loss | Margin $m$ adjustment based on the number of samples per class | None | $\max(0, m - \log(\text{prob}[c])) + \text{regularization}$ |
| PriorCELoss | The class prior distribution $\frac{n_c}{N}$ | None | $\text{logits} = x + \log(\text{prior})$
$\text{loss} = \text{CrossEntropy}(\text{logits}, y)$ |
| WeightedSoftmax | The normalized class probabilities | $\text{weight} = -\log(\text{normalized\_prob}) + 1$ | $\text{CE}(\text{logits}, y, \text{weight})$ |
| BalancedSoftmax | The number of samples per class, applied to logits | None | $\text{logits} = \log(\text{softmax}) - \log(\text{class\_distribution})$ |
| VSLoss | The number of samples per class with adjustments | Adjustment factors $\Delta$ and offsets $\iota$ applied to logits | $\text{CrossEntropy}\left(\frac{x_c}{\Delta_c} + \iota_c, y\right)$ |

as seen in MisLAS and GCL, is essential. The removal of MixUp from GCL in our experiments led to a significant performance decline, e.g, from 64.06% to 62.01% for ISIC-2019-LT with $r = 100$, underscoring its crucial role.

**Use two-stage training as a general paradigm.** Although end-to-end training is often considered to be elegant, addressing long-tailed problems undeniably requires distinct solutions at both the feature and classifier levels and two-stage methods provide significant flexibility in this regard. For feature representation, methods like mixup, as previously discussed, can enhance the representation learning, and self-supervised learning is another potential strategy to improve the generalization of feature encoders. At the classifier level, designing new classifiers can help correct the bias introduced by linear layers. We will delve into these aspects in more detail later.

**Dilemmas of self-supervised learning for LTMIC.** SSL appears promising for addressing long-tailed problems (Kang et al., 2021b; Cui et al., 2021), as it works without the need for label supervision, allowing the feature encoder to obtain more generalizable feature representation. We explore the general SSL methodologies such as MoCo (He et al., 2020), BYOL (Grill et al., 2020), SimCLR (Chen et al., 2020), and SSL for long-tailed learning such as SCL (Khosla et al., 2020), KCL (Kang et al., 2021a) and PaCo (Cui et al., 2021) but the performance is not satisfactory with catastrophic performance drop. We conclude this for several reasons: (1) The amount of medical imaging data in this study is limited, especially for a LTMIC setting, while SSL often relies on large quantities of unlabeled data. (2) SSL typically depends on specific hyper-parameter designs/tuning such as data augmentations, which are crucial to the second-stage fine-tuning. However, the lack of guidance in these aspects diminishes the possibility of achieving optimal practice. We have provided implementations of some SSL methodologies in our codebase for further modification or improvement by researchers.

**Modify classifier to reduce prediction bias.** The common practice for image classification is using a linear classifier, and the predictions can be formulated as $p = \text{Softmax}(WX+b)$. However, long-tailed class imbalance often results in larger classifier weight norms $W$ for head classes than tail classes, which makes the predictions easily biased to dominant classes. In Fig. 2, We visualize the trade-off between shot-based group performance and weight norms. This indicates that re-sampling strategies such as RW and CBLoss can further calibrate the classifier weight norms. GCL and DisAlign adopt a simple normalized linear classifier with learnable parameters to scale the predictions, leading to an optimal calibration of the weight norms. There are also some other compacted classifier designs (Kang et al., 2020; Tang et al., 2020). Considering that different classifier designs and other strategies, such as resampling methods, may conflict with each other, we still recommend starting with a simple design like a normalized linear classifier in practice.

**LTMIC improves out-of-distribution detection.** Open Long-tailed Recognition (OLTR) extends long-tailed learning by incorporating out-of-distribution (OOD) detection as an additional task. Models trained on imbalanced datasets typically struggle to generalize well to tail classes, making OLTR especially challenging due to the confusion between tail and OOD samples. To assess whether the LTMIC methods improve OOD detection capabilities, we used the OpenOOD codebase (Yang et al., 2022) and evaluated six in-built OOD detection methods. We used the model trained on the OrganAMNIST dataset, using its test set as closed-set samples and ImageNet (Deng et al., 2009) as OOD samples, each with 1,000 randomly selected images. AUROC was used as the evaluation metric for binary classification. As shown in Fig. 3, LTMIC leads to significant performance improvements across various OOD detection methods.

**Using imbalanced validation dataset for checkpoint selection.** Ideally, a balanced validation set would be preferred for selecting the best model checkpoint, as it ensures a fairer evaluation on the

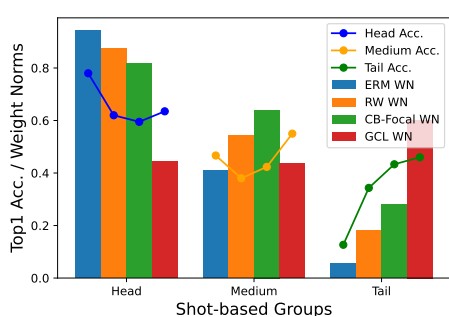 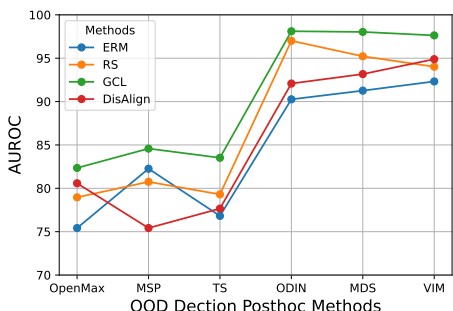

Figure 2: The performance and weight norms of the model trained from ISIC-2019-LT ($r$=500).

Figure 3: The performance of OOD detection methods with LTMIC methods.

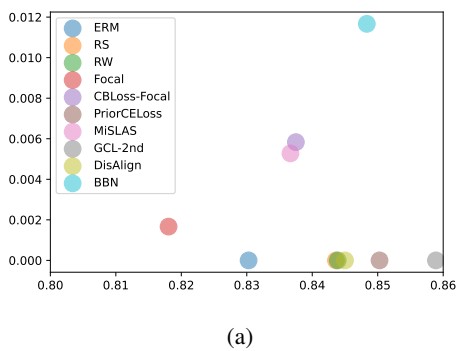

(a)

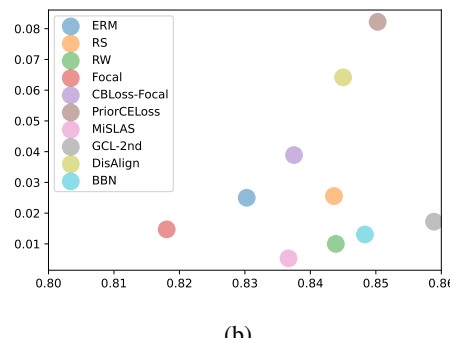

(b)

Figure 4: (a) Performance and the gap between the selected epoch (based on validation set performance) and the best test set epoch across different methods. (b) Performance and the gap between the selected epoch (based on validation set performance) and the final epoch across different methods.

test set. An imbalanced validation set may not fully represent the underlying data distribution, and in extreme cases, some tail categories could contain fewer than 10 samples. As a result, achieving high performance on the validation set may not translate to strong generalization on the test set or in real-world applications. Therefore, the method should demonstrate stability during training, with minimal performance fluctuations as it progresses. Without repeating trials, we conducted following two sets of experiments: (1) For each epoch, we evaluated both the validation and test sets. We then calculated the difference between the best test set performance and the test set performance at the epoch where the validation set achieved its highest performance. (2) We calculated the difference between the test set performance at the epoch with the best validation set performance and the test set performance at the final epoch. The results shown in Fig. 4 indicate that GCL demonstrates both strong overall performance and stable convergence during model training.

**Multi-label classification is more challenging.** Compared to multi-class (MC) classification, multi-label (ML) classification presents additional challenges, particularly due to label co-occurrence, where some labels frequently appear together (e.g., in medical imaging, "diabetic retinopathy" often co-occurs with "glaucoma"). This complicates the application of re-sampling strategies that are commonly used in MC problems with new relative imbalance introduced (Wu et al., 2020; Ju et al., 2023). Despite these complexities, many methods designed for MC tasks can be adapted for ML scenarios. Methods such as OLTR (Liu et al., 2019), RSKD (Ju et al., 2021), and HKGL (Ju et al., 2023) show consistent improvements over the standard ERM approach, as demonstrated in results from ODIR and RFMiD (Table 5, based on HKGL (Ju et al., 2023)). Notably, some methods like OLTR can achieve performance gains across head, medium, and tail classes simultaneously in ML tasks. This may be because improving the model's ability to detect positive samples in tail classes reduces overall misclassification of negative samples, thereby enhancing performance even for head classes. For MC tasks, we resampled datasets using a Pareto distribution to focus on the

Table 5: The results of comparison study on multi-label classification datasets. **\* -** denotes the methodology is not supported in MONICA.

| Dataset | RFMiD | | | | ODIR | | | | | | | | CheXpert | | | |
|---|---|---|---|---|---|---|---|---|---|---|---|---|---|---|---|---|
| Split | Test Set | | | | Off-Site | | | | On-Site | | | | Test Set | | | |
| Groups | Many | Medium | Few | Average | Many | Medium | few | average | many | medium | Few | Average | Many | Medium | Few | Average |
| ERM | 70.93 | 57.89 | 14.85 | 47.89 | 48.47 | 46.80 | 11.22 | 35.50 | 50.74 | 36.46 | 12.79 | 33.33 | **85.78** | **57.60** | 3.67 | 42.06 |
| RS | 68.67 | **61.48** | 25.94 | 52.03 | 46.34 | **49.27** | 9.07 | 34.89 | 47.91 | **39.10** | 15.35 | 34.12 | 73.28 | 48.76 | 13.65 | 45.23 |
| RW | 70.27 | 60.00 | 18.71 | 49.66 | **50.56** | 48.12 | 11.57 | 36.75 | 51.39 | 37.86 | 17.92 | 35.72 | 74.33 | 46.27 | **14.20** | 44.93 |
| OLTR | **71.25** | 60.22 | 20.77 | 50.75 | 47.37 | 45.02 | 11.86 | 34.75 | 50.11 | 36.01 | 20.78 | 35.63 | - | - | - | - |
| RSKD | 70.55 | 59.63 | 22.15 | 50.78 | 48.09 | 47.78 | 10.82 | 35.56 | 48.89 | 38.61 | **31.21** | **39.57** | - | - | - | - |
| Focal | 70.65 | 55.53 | 16.42 | 47.53 | 46.63 | 46.89 | 13.32 | 35.61 | 47.92 | 35.41 | 10.49 | 31.27 | 72.35 | 55.11 | 9.68 | **45.71** |
| LDAM | 46.67 | 3.19 | 1.18 | 17.01 | 41.14 | 8.22 | 0.48 | 16.61 | 42.97 | 5.10 | 0.55 | 16.21 | 63.63 | 10.14 | 3.29 | 25.58 |
| CBLoss-Focal | 67.73 | 50.89 | 24.65 | 47.77 | 39.30 | 47.44 | 10.00 | 32.25 | 43.40 | 32.31 | 8.60 | 28.10 | 65.83 | 10.45 | 3.00 | 26.30 |
| DBLoss-Focal | 68.16 | 55.27 | 18.94 | 47.46 | 48.39 | 47.11 | 27.83 | 41.11 | 50.06 | 37.60 | 12.96 | 33.54 | 74.44 | 46.12 | 13.95 | 44.83 |
| ASL | 68.25 | 58.25 | 19.59 | 48.70 | 47.93 | 47.89 | 18.57 | 38.13 | **51.69** | 37.36 | 23.70 | 37.58 | 73.86 | 45.08 | 13.50 | 44.15 |
| HKGL | 69.75 | 61.26 | **25.98** | **52.33** | 49.02 | 48.26 | **28.05** | **41.78** | 51.58 | 36.82 | 28.98 | 39.12 | - | - | - | - |

total number of data samples and the degree of imbalance. However, in ML tasks, the presence of label co-occurrence makes following a Pareto distribution impossible. This highlights the greater complexity of long-tailed multi-label learning compared to multi-class classification, where the interplay of factors such as imbalance ratio, label co-occurrence, and category distribution makes it challenging to draw unified conclusions or insights.

**What makes an essential strong baseline?** Our analysis shows that the most advanced long-tailed learning methods no longer focus on improving a single strategy. Instead, they integrate re-sampling, information augmentation, and module improvements, as exemplified by GCL. We would like to emphasize that LTMIC is primarily an engineering-focused effort. These include using more sophisticated augmentation techniques like RandAugment (Cubuk et al., 2020), employing models with larger parameters, and introducing various other tricks such as learning scheduler, attention mechanism, and knowledge distillation (Hinton et al., 2015). However, we have reservations about the results of these attempts to maintain focus on the core methods under investigation and avoid diluting the primary contributions of this study. Finally, given the inherent imperfections of long-tailed data, it is unrealistic to assume that a single method can deliver optimal performance across all categories. Therefore, it is necessary to consider trade-offs in performance between different shot-based groups and select methods based on specific needs.

**Integration of medical domain prior knowledge.** While this paper extensively explores state-of-the-art methods designed for natural image classification and validates their generalization across datasets from various medical domains, we still advocate for the integration of medical domain knowledge to develop specialized techniques, such as hierarchical learning (Ju et al., 2023), tailored to specific medical challenges. Incorporating prior knowledge helps guide the model to focus on critical features, thereby accelerating convergence and improving training efficiency, finally benefiting the overall performance. More importantly, the integration of clinical insights enhances model interpretability, enabling more transparent and clinically relevant decision-making, from which the importance and value of LTMIC research are exactly highlighted and appreciated.

## 5 CONCLUSION

In this study, we introduced MONICA, a comprehensive benchmark for long-tailed medical image classification (LTMIC). Our findings emphasize the importance of integrating techniques from multiple aspects including re-sampling, data augmentation, and module improvements, offering valuable practical guidance for future research in LTMIC. The modular design of our codebase further facilitates the application and comparison of these methods across various medical imaging tasks.

**Limitations and Future Works** This work is largely limited to the implementation of partial existing long-tailed learning works. Due to the unavailability of code, our implementation relies heavily on the details provided in the papers, which may lead to variations in performance. Another limitation is that multi-label learning, as a significant challenge within long-tailed learning, has distinct problem definitions, implementations, and evaluation metrics. However, this paper does not delve deeply into this aspect but instead provides a brief introduction and experimental results on some mainstream datasets. Finally, no novel metrics are proposed for better quantitative analysis. In future work, it is promising to extend our benchmark towards update works and more long-tailed learning tasks such as long-tailed multi-label learning, regression, object detection, and semantic segmentation.

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
