# A APPENDIX

Table 6: The complete comparison study results on ISIC-2019-LT benchmarks.

| | ISIC-2019-LT | | | | | | | | | | | |
|---|---|---|---|---|---|---|---|---|---|---|---|---|
| Imbalance Ratio | $r = 100$ | | | | $r = 200$ | | | | $r = 500$ | | | |
| Methods | Head | Medium | Tail | Average | Head | Medium | Tail | Average | Head | Medium | Tail | Average |
| ERM | 79.00 | 60.67 | 38.33 | 59.33 | 78.50 | 56.67 | 27.00 | 54.06 | 78.00 | 46.67 | 12.67 | 45.78 |
| RS | 69.50 | 61.33 | 49.33 | 60.06 | 76.00 | 62.67 | 36.33 | 58.33 | 78.50 | 44.00 | 19.67 | 47.39 |
| RW | 68.00 | 55.33 | 53.67 | 59.00 | 73.50 | 54.67 | 41.33 | 56.50 | 62.00 | 38.00 | 34.33 | 44.78 |
| MixUp | 78.00 | 50.67 | 35.67 | 54.78 | 83.00 | 46.67 | 21.33 | 50.33 | 76.00 | 48.00 | 9.00 | 44.33 |
| Focal | 73.50 | 54.00 | 44.33 | 57.28 | 79.50 | 53.00 | 31.00 | 54.50 | 83.00 | 44.00 | 13.00 | 46.67 |
| cRT | 74.50 | 59.67 | 55.67 | 63.28 | 81.00 | 60.00 | 39.67 | 60.22 | 63.50 | 48.33 | 28.00 | 46.61 |
| T-Norm | 78.00 | 53.33 | 34.00 | 55.11 | 83.50 | 48.67 | 17.67 | 49.94 | 79.50 | 40.33 | 3.67 | 41.17 |
| LWS | 72.50 | 52.67 | 45.33 | 56.83 | 79.50 | 51.33 | 32.67 | 54.50 | 68.00 | 43.33 | 29.67 | 47.00 |
| KNN | 70.00 | 55.67 | 58.67 | 61.44 | 77.00 | 51.33 | 46.67 | 58.33 | 75.00 | 45.33 | 27.33 | 49.22 |
| CBLoss | 70.00 | 56.33 | 61.33 | 62.56 | 62.00 | 59.33 | 46.67 | 56.00 | 55.00 | 40.33 | 41.67 | 45.67 |
| CBLoss_Focal | 71.00 | 57.67 | 52.67 | 60.44 | 72.50 | 52.00 | 51.00 | 58.50 | 59.50 | 42.33 | 43.33 | 48.39 |
| LADELoss | 78.50 | 52.33 | 43.67 | 58.17 | 84.00 | 52.33 | 18.67 | 51.67 | 78.50 | 43.00 | 14.00 | 45.17 |
| LDAM | 78.50 | 55.67 | 41.67 | 58.61 | 81.50 | 52.33 | 31.00 | 54.94 | 76.50 | 41.33 | 19.33 | 45.72 |
| Logits Adjust Loss | 80.50 | 50.67 | 26.33 | 52.50 | 81.00 | 49.33 | 12.67 | 47.67 | 77.50 | 36.00 | 2.67 | 38.72 |
| Logits Adjust Posthoc | 75.00 | 59.33 | 49.33 | 61.22 | 76.50 | 50.67 | 48.00 | 58.39 | 72.00 | 43.67 | 33.00 | 49.56 |
| PriorCELoss | 65.00 | 57.00 | 56.00 | 59.33 | 75.00 | 50.33 | 48.33 | 57.89 | 62.00 | 52.67 | 43.33 | 52.67 |
| RangeLoss | 29.50 | 12.00 | 0.33 | 13.94 | 44.50 | 9.00 | 4.00 | 19.17 | 46.00 | 5.67 | 0.67 | 17.44 |
| SEQLLoss | 78.00 | 56.33 | 41.00 | 58.44 | 78.00 | 49.67 | 27.00 | 51.56 | 73.50 | 45.33 | 11.33 | 43.39 |
| VSLoss | 80.00 | 56.33 | 33.67 | 56.67 | 80.50 | 51.00 | 28.67 | 53.39 | 79.00 | 47.67 | 11.00 | 45.89 |
| WeightedSoftmax | 70.50 | 58.00 | 48.33 | 58.94 | 74.00 | 59.33 | 36.00 | 56.44 | 73.00 | 48.67 | 22.00 | 47.89 |
| BalancedSoftmax | 62.50 | 54.33 | 61.00 | 59.28 | 77.00 | 53.67 | 55.00 | 61.89 | 62.00 | 49.67 | 41.67 | 51.11 |
| De-Confound | 79.00 | 52.33 | 47.67 | 59.67 | 82.50 | 52.67 | 24.33 | 53.17 | 72.50 | 45.33 | 9.67 | 42.50 |
| DisAlign | 81.00 | 60.00 | 52.33 | 64.44 | 78.00 | 59.67 | 52.33 | 63.33 | 68.50 | 49.33 | 37.33 | 51.72 |
| GCL 1st stage | 72.00 | 56.00 | 50.33 | 59.44 | 78.50 | 55.33 | 32.67 | 55.50 | 78.00 | 41.33 | 31.67 | 50.33 |
| GCL 2nd stage | 57.50 | 63.33 | 71.33 | 64.06 | 71.00 | 56.67 | 64.33 | 64.00 | 63.50 | 55.00 | 46.00 | 54.83 |
| MiSLAS | 57.50 | 52.33 | 57.67 | 55.83 | 71.50 | 48.33 | 49.67 | 56.50 | 63.50 | 43.00 | 39.33 | 48.61 |
| RSG | 72.00 | 35.00 | 0.00 | 35.67 | 78.50 | 23.33 | 0.00 | 33.94 | 76.00 | 23.67 | 0.00 | 33.22 |
| SADE | 34.50 | 19.33 | 48.00 | 33.94 | 31.50 | 11.33 | 52.67 | 31.83 | 23.50 | 13.00 | 48.33 | 28.28 |
| SAM | 77.50 | 51.67 | 26.00 | 51.72 | 82.50 | 50.33 | 16.33 | 49.72 | 76.00 | 36.00 | 9.33 | 40.44 |
| BBN | 82.50 | 58.33 | 46.00 | 62.28 | 76.50 | 62.33 | 31.00 | 54.94 | 75.50 | 50.67 | 19.00 | 48.39 |

Table 7: The results on ISIC-2019-LT benchmarks with training epochs of 50 and 200.

| | ISIC-2019-LT ($r = 100$) | | | | | | | |
|---|---|---|---|---|---|---|---|---|
| Training Epochs | 50 | | | | 200 | | | |
| Methods | Head | Medium | Tail | Average | Head | Medium | Tail | Average |
| ERM | 79.00 | 60.67 | 38.33 | 59.33 | 78.50 | 54.00 | 45.00 | 59.17 |
| RS | 69.50 | 61.33 | 49.33 | 60.06 | 74.50 | 59.00 | 53.67 | 62.39 |
| RW | 68.00 | 55.33 | 53.67 | 59.00 | 78.50 | 59.00 | 51.00 | 62.83 |
| MixUp | 78.00 | 50.67 | 35.67 | 54.78 | 82.50 | 53.67 | 42.33 | 59.50 |
| Focal | 73.50 | 54.00 | 44.33 | 57.28 | 77.00 | 60.67 | 43.00 | 60.22 |
| cRT | 74.50 | 59.67 | 55.67 | 63.28 | 75.50 | 53.67 | 55.67 | 61.61 |
| T-Norm | 78.00 | 53.33 | 34.00 | 55.11 | 82.50 | 51.67 | 35.67 | 56.61 |
| LWS | 72.50 | 52.67 | 45.33 | 56.83 | 76.50 | 35.00 | 6.67 | 39.39 |
| KNN | 70.00 | 55.67 | 58.67 | 61.44 | 73.00 | 55.33 | 53.67 | 60.67 |
| CBLoss | 70.00 | 56.33 | 61.33 | 62.56 | 65.50 | 56.00 | 61.33 | 60.94 |
| CBLoss_Focal | 71.00 | 57.67 | 52.67 | 60.44 | 72.50 | 55.67 | 54.33 | 60.83 |
| LADELoss | 78.50 | 52.33 | 43.67 | 58.17 | 81.00 | 58.00 | 44.33 | 61.11 |
| LDAM | 78.50 | 55.67 | 41.67 | 58.61 | 74.50 | 59.00 | 40.00 | 57.83 |
| Logits Adjust Loss | 80.50 | 50.67 | 26.33 | 52.50 | 77.00 | 54.33 | 32.67 | 54.67 |
| Logits Adjust Posthoc | 75.00 | 59.33 | 49.33 | 61.22 | 76.00 | 59.00 | 53.00 | 62.67 |
| PriorCELoss | 65.00 | 57.00 | 56.00 | 59.33 | 73.00 | 58.33 | 56.67 | 62.67 |
| RangeLoss | 29.50 | 12.00 | 0.33 | 13.94 | 50.00 | 0.00 | 0.00 | 16.67 |
| SEQLLoss | 78.00 | 56.33 | 41.00 | 58.44 | 77.00 | 55.67 | 46.00 | 59.56 |
| VSLoss | 80.00 | 56.33 | 33.67 | 56.67 | 80.50 | 59.00 | 41.67 | 60.39 |
| WeightedSoftmax | 70.50 | 58.00 | 48.33 | 58.94 | 76.50 | 52.33 | 48.67 | 59.17 |
| BalancedSoftmax | 62.50 | 54.33 | 61.00 | 59.28 | 75.00 | 59.00 | 49.67 | 61.22 |
| De-Confound | 79.00 | 52.33 | 47.67 | 59.67 | 81.00 | 55.67 | 47.67 | 61.44 |
| DisAlign | 81.00 | 60.00 | 52.33 | 64.44 | 65.50 | 58.67 | 60.67 | 61.61 |
| GCL 1st stage | 72.00 | 56.00 | 50.33 | 59.44 | 78.50 | 59.00 | 56.33 | 64.61 |
| GCL 2nd stage | 57.50 | 63.33 | 71.33 | 64.06 | 72.50 | 63.67 | 63.67 | 66.61 |
| MiSLAS | 57.50 | 52.33 | 57.67 | 55.83 | 62.00 | 57.00 | 58.33 | 59.11 |
| RSG | 72.00 | 35.00 | 0.00 | 35.67 | 72.00 | 60.00 | 51.00 | 61.00 |
| SADE | 34.50 | 19.33 | 48.00 | 33.94 | 65.00 | 49.00 | 47.00 | 53.67 |
| SAM | 77.50 | 51.67 | 26.00 | 51.72 | 80.50 | 50.33 | 33.33 | 54.72 |
| BBN | 82.50 | 58.33 | 46.00 | 62.28 | 74.00 | 61.00 | 53.67 | 62.89 |

**How training time impacts method effectiveness.** Table 7 presents the results for ISIC-2019-LT ($r = 100$) across different methods trained with 50 and 200 epochs. We observed that some methods benefit significantly from extended training; for example, RSG improves from 35.67% AUC at epoch 50 to 61.00% AUC at epoch 200. Conversely, methods like DisAlign exhibit performance degradation with longer training. This suggests potential instability during training and highlights the issue of using an imbalanced validation set to select checkpoints for testing, as discussed in Section 4 of the manuscript. Overall, most methods do not reach optimal performance within 50 epochs, and longer training times may be beneficial for further improvements. However, it is important to emphasize that different methods are designed with varying requirements for learning rates and optimizers depending on different datasets. As such, conclusions based solely on these results may be partial. These findings are intended to offer researchers and engineers guidance on tuning hyperparameters when training with their customized data.

**The importance of pre-trained weights initialization.** Using ImageNet pre-trained weights as initialization weights is highly beneficial for medical image classification. Despite the huge differences in overall style between domains, ImageNet pretraining offers a common coarse learning representation of texture features and provides a good initialization for optimization convergence. Thus, its benefits will not be further elaborated here. Table 8 presents the performance differences across various methods when using or not using ImageNet for weight initialization. It is worth noting that some methods have modified model structures, making it difficult for these models to leverage ImageNet weights for initialization. However, we would like to emphasize the importance of using well-pretrained models **in practice**, while the results in the table further corroborate this point.

Table 8: The results on ISIC-2019-LT benchmarks w/ and w/o ImageNet pre-training ($r = 100$).

| ISIC-2019-LT ($r = 100$) | | | | | | | |
|---|---|---|---|---|---|---|---|
| Initialized Weights | ImageNet-1K | | | | Random | | |
| **Methods** | **Head** | **Medium** | **Tail** | **Average** | **Head** | **Medium** | **Tail** | **Average** |
| ERM | 79.00 | 60.67 | 38.33 | 59.33 | 73.00 | 21.33 | 0.33 | 31.56 |
| RS | 69.50 | 61.33 | 49.33 | 60.06 | 56.50 | 33.00 | 56.00 | 48.50 |
| RW | 68.00 | 55.33 | 53.67 | 59.00 | 53.00 | 24.33 | 36.67 | 38.00 |
| MixUp | 78.00 | 50.67 | 35.67 | 54.78 | 68.50 | 19.67 | 0.67 | 29.61 |
| Focal | 73.50 | 54.00 | 44.33 | 57.28 | 67.00 | 27.67 | 1.00 | 31.89 |
| cRT | 74.50 | 59.67 | 55.67 | 63.28 | 51.50 | 31.33 | 61.67 | 48.17 |
| T-Norm | 78.00 | 53.33 | 34.00 | 55.11 | 73.50 | 20.33 | 0.00 | 31.28 |
| LWS | 72.50 | 52.67 | 45.33 | 56.83 | 66.00 | 7.00 | 0.00 | 24.33 |
| KNN | 70.00 | 55.67 | 58.67 | 61.44 | 56.50 | 26.00 | 33.00 | 38.50 |
| CBLoss | 70.00 | 56.33 | 61.33 | 62.56 | 46.00 | 26.33 | 31.00 | 34.44 |
| CBLoss_Focal | 71.00 | 57.67 | 52.67 | 60.44 | 48.50 | 34.00 | 26.33 | 36.28 |
| LADELoss | 78.50 | 52.33 | 43.67 | 58.17 | 72.00 | 23.00 | 0.00 | 31.67 |
| LDAM | 78.50 | 55.67 | 41.67 | 58.61 | 71.00 | 19.00 | 0.00 | 30.00 |
| Logits Adjust Loss | 80.50 | 50.67 | 26.33 | 52.50 | 70.50 | 16.67 | 0.00 | 29.06 |
| Logits Adjust Posthoc | 75.00 | 59.33 | 49.33 | 61.22 | 58.50 | 29.33 | 33.33 | 40.39 |
| PriorCELoss | 65.00 | 57.00 | 56.00 | 59.33 | 62.00 | 28.67 | 39.33 | 43.33 |
| RangeLoss | 29.50 | 12.00 | 0.33 | 13.94 | 40.00 | 12.33 | 0.00 | 17.44 |
| SEQLLoss | 78.00 | 56.33 | 41.00 | 58.44 | 73.50 | 26.67 | 0.67 | 33.61 |
| VSLoss | 80.00 | 56.33 | 33.67 | 56.67 | 73.00 | 20.33 | 0.00 | 31.11 |
| WeightedSoftmax | 70.50 | 58.00 | 48.33 | 58.94 | 63.50 | 30.00 | 10.00 | 34.50 |
| BalancedSoftmax | 62.50 | 54.33 | 61.00 | 59.28 | 64.00 | 29.67 | 29.67 | 41.11 |
| De-Confound | 79.00 | 52.33 | 47.67 | 59.67 | 71.50 | 20.33 | 0.33 | 30.72 |
| DisAlign | 81.00 | 60.00 | 52.33 | 64.44 | 50.00 | 27.67 | 49.67 | 42.44 |
| GCL 1st stage | 72.00 | 56.00 | 50.33 | 59.44 | 65.00 | 21.33 | 0.00 | 28.78 |
| GCL 2nd stage | 57.50 | 63.33 | 71.33 | 64.06 | 30.00 | 12.33 | 64.00 | 35.44 |
| MiSLAS | 57.50 | 52.33 | 57.67 | 55.83 | 42.00 | 14.67 | 35.00 | 30.56 |
| RSG | 72.00 | 35.00 | 0.00 | 35.67 | 63.00 | 50.67 | 54.33 | 56.00 |
| SADE | 34.50 | 19.33 | 48.00 | 33.94 | 40.00 | 19.00 | 50.33 | 36.44 |
| SAM | 77.50 | 51.67 | 26.00 | 51.72 | 53.50 | 13.33 | 0.00 | 22.28 |
| BBN | 82.50 | 58.33 | 46.00 | 62.28 | 46.30 | 46.67 | 42.74 | 45.24 |

**Additional details for OOD detection.** To assess whether the LTMIC methods improve OOD detection capabilities, we used the OpenOOD codebase (Yang et al., 2022) and evaluated six in-built OOD detection methods. We used the model trained on the Organamnist dataset, using its test set as closed-set samples and ImageNet as OOD samples, each with 1,000 randomly selected images. This task mainly aims to distinguish whether the test sample belongs to known classes from OrganAMNIST or unknown classes from ImageNet. AUROC was used as the evaluation metric for this binary classification. To avoid the impact of retraining models on long-tailed recognition task performance, we selected six post-processing methods, which are as follows: OpenMax (Bendale & Boult, 2016), MSP (Hendrycks & Gimpel, 2016), TS (Guo et al., 2017), ODIN (Liang et al., 2017),

MDS (Lee et al., 2018), and VIM (Wang et al., 2022). We used 0.01 for coreset sampling ratio in OpenMax. For ODIN, temperature is set as 1000 and noise is set as 0.0014.

Table 9: The complete results on MedMNIST and KVASIR benchmarks.

| Dataset | BloodMNIST | | | | DermaMNIST | | | | PathMNIST | | | | TissueMNIST | | | |
|---|---|---|---|---|---|---|---|---|---|---|---|---|---|---|---|---|
| **Methods** | **Head** | **Medium** | **Tail** | **Average** | **Head** | **Medium** | **Tail** | **Average** | **Head** | **Medium** | **Tail** | **Average** | **Head** | **Medium** | **Tail** | **Average** |
| ERM | 97.27 | 97.16 | 82.00 | 92.14 | 95.82 | 61.86 | 44.08 | 67.25 | 98.46 | 99.64 | 83.07 | 93.72 | 64.38 | 62.11 | 23.21 | 49.90 |
| RS | 95.75 | 99.20 | 91.17 | 95.37 | 91.95 | 65.75 | 59.75 | 72.48 | 99.26 | 97.47 | 86.78 | 94.50 | 50.26 | 64.71 | 52.88 | 55.95 |
| RW | 97.39 | 99.04 | 88.14 | 94.86 | 87.32 | 65.31 | 69.19 | 73.94 | 96.62 | 99.74 | 88.57 | 94.97 | 59.72 | 66.15 | 43.44 | 56.44 |
| MixUp | 98.04 | 98.93 | 79.93 | 92.30 | 95.90 | 59.03 | 54.43 | 69.78 | 99.38 | 99.54 | 84.72 | 94.54 | 65.85 | 62.79 | 13.54 | 47.39 |
| Focal | 97.09 | 99.42 | 80.75 | 92.42 | 95.08 | 59.60 | 60.57 | 71.75 | 96.38 | 98.66 | 87.51 | 94.18 | 61.89 | 63.17 | 22.07 | 49.04 |
| cRT | 97.64 | 99.36 | 87.43 | 94.81 | 88.52 | 73.36 | 70.09 | 77.32 | 93.86 | 99.69 | 90.59 | 94.71 | 59.47 | 67.83 | 38.70 | 55.33 |
| T-Norm | 96.45 | 97.97 | 80.79 | 91.74 | 95.97 | 56.76 | 44.16 | 65.63 | 99.47 | 97.12 | 85.51 | 94.03 | 66.57 | 52.56 | 9.82 | 42.98 |
| LWS | 90.30 | 75.36 | 1.50 | 55.72 | 95.90 | 25.97 | 1.73 | 41.20 | 98.41 | 98.30 | 52.25 | 82.98 | 59.28 | 33.59 | 0.60 | 31.16 |
| KNN | 93.61 | 95.88 | 83.31 | 90.94 | 85.31 | 66.62 | 59.67 | 70.53 | 93.56 | 99.79 | 89.67 | 94.34 | 54.77 | 55.40 | 51.38 | 53.85 |
| CBLoss | 96.92 | 98.23 | 85.81 | 93.65 | 83.15 | 63.57 | 76.61 | 74.44 | 97.51 | 99.64 | 87.69 | 94.94 | 60.38 | 67.61 | 43.66 | 57.21 |
| CBLoss_Focal | 96.97 | 98.34 | 85.21 | 93.51 | 84.41 | 63.27 | 73.99 | 73.89 | 98.44 | 99.45 | 84.10 | 94.00 | 58.60 | 67.08 | 45.31 | 56.99 |
| LADELoss | 97.16 | 98.77 | 72.76 | 89.56 | 91.28 | 67.43 | 50.60 | 69.77 | 98.55 | 98.48 | 84.43 | 93.82 | 65.96 | 63.33 | 23.92 | 51.07 |
| LDAM | 96.74 | 98.45 | 87.32 | 94.17 | 92.62 | 63.71 | 53.15 | 69.82 | 97.37 | 98.66 | 86.07 | 94.03 | 63.31 | 64.57 | 14.64 | 47.51 |
| Logits Adjust Loss | 95.41 | 98.18 | 51.39 | 81.66 | 97.39 | 52.12 | 43.63 | 64.38 | 99.58 | 98.58 | 75.68 | 91.28 | 59.79 | 44.62 | 1.23 | 35.21 |
| Logits Adjust Posthoc | 96.65 | 97.91 | 90.27 | 94.94 | 92.62 | 64.53 | 61.47 | 72.87 | 98.07 | 99.17 | 86.19 | 94.48 | 59.03 | 66.35 | 54.78 | 60.06 |
| PriorCELoss | 96.86 | 98.02 | 91.90 | 95.59 | 77.63 | 72.80 | 70.54 | 73.66 | 98.51 | 97.01 | 88.97 | 94.83 | 55.73 | 69.21 | 51.54 | 58.83 |
| RangeLoss | 23.58 | 6.11 | 3.69 | 11.13 | 100.00 | 0.00 | 0.00 | 33.33 | 50.00 | 0.00 | 0.00 | 16.67 | 33.34 | 0.00 | 0.00 | 11.11 |
| SEQLLoss | 97.22 | 97.27 | 85.27 | 93.26 | 89.93 | 69.13 | 55.77 | 71.61 | 97.53 | 99.59 | 88.21 | 95.11 | 65.90 | 62.00 | 24.12 | 50.68 |
| VSLoss | 96.05 | 97.70 | 86.59 | 93.45 | 92.84 | 64.75 | 49.70 | 69.10 | 98.87 | 98.31 | 83.46 | 93.54 | 66.78 | 61.45 | 21.54 | 49.92 |
| WeightedSoftmax | 97.78 | 98.98 | 84.65 | 93.80 | 87.84 | 68.31 | 69.27 | 75.14 | 98.83 | 95.11 | 85.41 | 93.12 | 65.38 | 68.36 | 35.57 | 56.43 |
| BalancedSoftmax | 96.55 | 98.13 | 90.71 | 95.13 | 89.41 | 67.51 | 76.61 | 77.79 | 96.39 | 99.54 | 88.42 | 94.78 | 53.86 | 68.41 | 52.91 | 58.39 |
| De-Confound | 97.14 | 97.49 | 85.10 | 93.24 | 95.08 | 61.54 | 49.25 | 68.62 | 93.45 | 99.85 | 88.53 | 93.94 | 65.32 | 61.35 | 24.34 | 50.34 |
| DisAlign | 97.11 | 98.98 | 88.82 | 94.97 | 85.68 | 66.04 | 75.72 | 75.81 | 96.70 | 97.95 | 91.85 | 95.50 | 50.20 | 59.73 | 60.21 | 56.71 |
| GCL 1st stage | 96.98 | 99.52 | 90.70 | 95.74 | 88.07 | 65.34 | 57.95 | 70.45 | 98.41 | 98.30 | 87.80 | 94.62 | 64.44 | 69.99 | 33.69 | 56.04 |
| GCL 2nd stage | 96.43 | 99.47 | 95.56 | 97.15 | 68.75 | 73.11 | 90.03 | 77.30 | 93.00 | 99.58 | 91.79 | 94.79 | 48.26 | 68.48 | 46.93 | 54.56 |
| MiSLAS | 93.15 | 99.25 | 81.74 | 91.38 | 65.92 | 67.53 | 75.72 | 69.72 | 96.30 | 98.99 | 88.63 | 94.64 | 42.84 | 64.71 | 49.91 | 52.49 |
| RSG | 95.45 | 98.56 | 86.55 | 93.52 | 77.33 | 53.90 | 63.12 | 64.78 | 96.16 | 99.95 | 70.92 | 89.01 | 53.08 | 59.44 | 0.00 | 37.51 |
| SADE | 60.99 | 81.91 | 52.37 | 65.09 | 59.36 | 30.89 | 58.47 | 49.57 | 69.45 | 95.87 | 76.96 | 80.76 | 45.23 | 59.76 | 58.09 | 54.36 |
| SAM | 96.10 | 97.49 | 77.40 | 90.33 | 93.36 | 61.02 | 36.66 | 63.68 | 99.30 | 99.64 | 86.89 | 95.28 | 64.28 | 69.99 | 46.93 | 54.56 |
| BBN | 96.24 | 98.23 | 89.39 | 94.62 | 91.87 | 70.98 | 64.02 | 75.62 | 98.14 | 98.95 | 90.33 | 95.80 | 58.62 | 68.41 | 43.29 | 57.19 |

| Dataset | OrganAMNIST | | | | OrganCMNIST | | | | OrganSMNIST | | | | KVASIR | | | |
|---|---|---|---|---|---|---|---|---|---|---|---|---|---|---|---|---|
| **Methods** | **Head** | **Medium** | **Tail** | **Average** | **Head** | **Medium** | **Tail** | **Average** | **Head** | **Medium** | **Tail** | **Average** | **Head** | **Medium** | **Tail** | **Average** |
| ERM | 97.27 | 97.16 | 82.00 | 92.14 | 93.50 | 59.20 | 65.18 | 72.63 | 88.33 | 56.89 | 66.90 | 70.71 | 95.50 | 93.25 | 60.33 | 83.03 |
| RS | 95.75 | 99.20 | 91.17 | 95.37 | 92.49 | 57.14 | 64.06 | 71.23 | 89.99 | 52.61 | 66.39 | 69.66 | 95.75 | 94.50 | 62.83 | 84.36 |
| RW | 97.39 | 99.04 | 88.14 | 94.86 | 92.20 | 68.51 | 66.84 | 75.98 | 83.01 | 58.68 | 69.60 | 70.43 | 96.75 | 88.75 | 67.67 | 84.39 |
| MixUp | 98.04 | 98.93 | 79.93 | 92.30 | 92.25 | 57.43 | 66.97 | 72.22 | 89.15 | 50.09 | 64.25 | 67.83 | 96.25 | 92.25 | 55.33 | 81.28 |
| Focal | 97.09 | 99.42 | 80.75 | 92.42 | 91.98 | 57.02 | 63.36 | 70.79 | 86.24 | 54.93 | 64.77 | 68.65 | 95.75 | 88.00 | 61.83 | 81.81 |
| cRT | 97.64 | 99.36 | 87.43 | 94.81 | 92.11 | 63.17 | 67.38 | 74.22 | 88.23 | 58.99 | 69.79 | 72.34 | 95.50 | 88.00 | 68.83 | 84.11 |
| T-Norm | 96.45 | 97.97 | 80.79 | 91.74 | 92.17 | 55.15 | 63.93 | 70.42 | 86.82 | 54.09 | 66.73 | 68.92 | 98.00 | 89.75 | 59.50 | 82.42 |
| LWS | 90.30 | 75.36 | 1.50 | 55.72 | 44.87 | 7.82 | 1.26 | 17.98 | 59.44 | 32.10 | 23.94 | 38.49 | 95.00 | 87.00 | 61.67 | 81.22 |
| KNN | 93.63 | 95.88 | 83.31 | 90.94 | 88.66 | 56.54 | 65.97 | 70.39 | 85.33 | 54.82 | 66.97 | 69.04 | 95.00 | 91.50 | 67.00 | 84.50 |
| CBLoss | 96.92 | 98.23 | 85.81 | 93.65 | 94.21 | 58.69 | 66.39 | 73.10 | 79.87 | 65.08 | 73.91 | 72.96 | 93.50 | 90.25 | 71.00 | 84.92 |
| CBLoss_Focal | 96.97 | 98.34 | 85.21 | 93.51 | 94.16 | 57.99 | 67.36 | 73.17 | 84.18 | 60.42 | 71.42 | 72.00 | 95.00 | 84.75 | 71.50 | 83.75 |
| LADELoss | 97.16 | 98.77 | 72.76 | 89.56 | 91.22 | 61.58 | 62.86 | 71.88 | 83.32 | 59.04 | 64.92 | 69.09 | 96.25 | 90.00 | 60.83 | 82.36 |
| LDAM | 96.74 | 98.45 | 87.32 | 94.17 | 90.98 | 61.68 | 69.16 | 73.94 | 86.07 | 58.66 | 68.60 | 71.11 | 96.00 | 89.00 | 63.17 | 82.72 |
| Logits Adjust Loss | 95.41 | 98.18 | 51.39 | 81.66 | 91.37 | 55.47 | 56.10 | 67.65 | 89.31 | 53.47 | 56.78 | 66.52 | 96.75 | 93.50 | 48.17 | 79.47 |
| Logits Adjust Posthoc | 96.65 | 97.91 | 90.27 | 94.94 | 91.75 | 62.72 | 71.24 | 75.24 | 82.78 | 62.01 | 69.10 | 71.29 | 93.00 | 89.75 | 70.67 | 84.47 |
| PriorCELoss | 96.86 | 98.02 | 91.90 | 95.59 | 92.83 | 64.86 | 67.17 | 74.95 | 86.08 | 60.96 | 72.36 | 73.13 | 95.50 | 89.25 | 70.33 | 85.03 |
| RangeLoss | 23.58 | 6.11 | 3.69 | 11.13 | 0.00 | 33.33 | 0.00 | 11.11 | 0.00 | 33.33 | 0.00 | 11.11 | 0.00 | 25.50 | 0.00 | 8.50 |
| SEQLLoss | 97.22 | 97.27 | 85.27 | 93.26 | 94.52 | 61.45 | 65.15 | 73.71 | 88.83 | 55.61 | 63.84 | 69.43 | 97.50 | 90.50 | 58.00 | 82.00 |
| VSLoss | 96.05 | 97.70 | 86.59 | 93.45 | 94.79 | 61.42 | 67.08 | 74.43 | 86.46 | 58.66 | 66.70 | 70.61 | 96.75 | 92.75 | 62.83 | 84.11 |
| WeightedSoftmax | 97.78 | 98.98 | 84.65 | 93.80 | 95.06 | 65.70 | 69.36 | 76.70 | 84.02 | 60.64 | 67.67 | 70.77 | 95.25 | 93.25 | 68.17 | 85.56 |
| BalancedSoftmax | 96.55 | 98.13 | 90.71 | 95.13 | 93.58 | 62.68 | 67.61 | 74.62 | 81.44 | 59.27 | 70.05 | 70.26 | 95.25 | 88.25 | 65.83 | 83.11 |
| De-Confound | 97.14 | 97.49 | 85.10 | 93.24 | 92.56 | 62.81 | 64.58 | 73.98 | 87.03 | 61.35 | 66.07 | 71.48 | 96.25 | 90.50 | 66.50 | 84.42 |
| DisAlign | 97.11 | 98.98 | 88.82 | 94.97 | 93.74 | 62.69 | 72.65 | 76.36 | 88.09 | 58.77 | 66.50 | 71.12 | 95.75 | 89.25 | 68.50 | 84.50 |
| GCL 1st stage | 96.98 | 99.52 | 90.70 | 95.74 | 94.15 | 60.46 | 68.99 | 74.54 | 88.64 | 62.68 | 77.80 | 74.80 | 96.00 | 93.75 | 65.67 | 85.14 |
| GCL 2nd stage | 96.43 | 99.47 | 95.56 | 97.15 | 92.99 | 64.40 | 75.42 | 77.60 | 81.80 | 66.23 | 76.79 | 74.94 | 95.25 | 96.25 | 66.17 | 85.89 |
| MiSLAS | 93.15 | 99.25 | 81.74 | 91.38 | 84.86 | 51.08 | 64.40 | 66.78 | 74.66 | 52.74 | 69.39 | 65.60 | 94.25 | 85.75 | 71.00 | 83.67 |
| RSG | 95.45 | 98.56 | 86.55 | 93.52 | 86.48 | 52.89 | 64.18 | 67.85 | 82.77 | 66.06 | 70.06 | 72.96 | 95.50 | 91.50 | 65.33 | 84.11 |
| SADE | 60.99 | 81.91 | 52.37 | 65.09 | 67.88 | 46.28 | 52.30 | 55.49 | 72.74 | 48.62 | 59.27 | 60.21 | 85.75 | 77.00 | 43.50 | 68.75 |
| SAM | 96.10 | 97.49 | 77.40 | 90.33 | 91.34 | 59.89 | 64.26 | 71.83 | 90.77 | 56.11 | 63.36 | 70.08 | 97.00 | 93.25 | 60.50 | 83.58 |
| BBN | 96.24 | 98.23 | 89.39 | 94.62 | 91.55 | 63.41 | 63.86 | 72.94 | 87.91 | 59.28 | 71.74 | 72.98 | 95.25 | 89.25 | 70.00 | 84.83 |