# OpenReview forum: "MONICA: Benchmarking on Long-tailed Medical Image Classification"
_ICLR.cc/2025/Conference — Submitted to ICLR 2025_

### Official Review · Reviewer_FbkS · 2024-10-18

**Soundness:** 4
**Presentation:** 3
**Contribution:** 3
**Rating:** 8
**Confidence:** 4

**Summary:**

This paper presents a framework and codebase for structured benchmarking of long-tailed (LT) learning methods on various medical image classification tasks. The benchmark, MONICA, implements over 30 LT learning methods, with comprehensive experiments assessing performance across 12 LT medical image classification datasets spanning 6 different modalities. The experiments analyze which methods, and categories of methods, provide the most benefit to LT medical image classification across tasks in a controlled environment.

**Strengths:**

- This work addresses an important problem, namely the variability in dataset/hyperparameters/etc. when evaluating LT learning methods for medical image classification tasks. These variations in setting make head-to-head comparisons difficult, so MONICA serves to provide a “fair playing ground” for these LT learning methods.
- The framework will become publicly available and should serve as an extensible resource going forward for LT medical image classification research.
- The organization and presentation quality of the paper is strong, with helpful use of formatting (typesetting, color, etc.) and high-quality figures.
- Experiments are very thorough, spanning many relevant methods, datasets, and tasks.
- Discussion is thoughtful, going beyond simply displaying all benchmark results. The authors try to synthesize takeaways, provide caveats/limitations, assess out-of-distribution performance, and more.

**Weaknesses:**

- Writing can be improved throughout. See specific comments below for examples of awkward wording, inconsistent naming, grammatical errors, etc.
- It is possible that choosing a fixed set of hyperparameters across methods unintentionally advantages certain methods. Ideally, one could argue that each method should be individually tuned on each task; however, I am aware that this would require a vast amount of resources and time, so I do not consider this a major limitation. More practical solutions to enhance the benchmark would be the following: (i) uncertainty estimates should be provided (e.g., bootstrapped confidence intervals or standard deviations over multiple runs), and (ii) multiple performance metrics should be provided (e.g., AUROC).

**Questions:**

- Is it possible that the chosen hyperparameters used across all methods happen to be more advantageous for certain methods and suboptimal for others? In one sense, using the same set of hyperparameters across methods appears “fair”; however, it may actually be more fair to individually tune each method on each task. I recognize the difficulty of conducting fair comparisons in such a large-scale experimental setting, where it is costly to, e.g., run multiple trials of all experiments. I am not asking the authors to necessarily perform such experiments, but rather to consider this point and perhaps comment on it as a limitation/consideration.
- Can the authors provide a summary of practical suggestions for which methods to use in a few sentences near the Conclusion?
- I might suggest including the **rank** of each method on a given task in all tables. This would also enable you to *quantitatively* assess method performance across tasks (which method has the lowest average/median rank overall?). To make this work logistically (fit all columns in the table), you may need to reduce the precision to one decimal place, e.g.
- The two paragraphs “LTMIC improves out-of-distribution detection” and “Using imbalanced validation dataset for checkpoint selection” are not properly set up. For the former, what does it mean to use “ImageNet as OOD samples, with 1,000 randomly selected images”? What exactly is the task, how is it formulated, and how are experiments conducted? Further, why do we care about this model behavior? For the former, Figure 4 and its findings are confusing – why exactly does this demonstrate “stable convergence”? My general advice: **Use the methods section to describe and prepare the reader to understand everything that appears in the results**. When I come to these results sections, I should already have an idea of what experiments you have performed.

**Minor comments/questions:**
- Avoid editorializing with value judgments: “benchmark is **meticulously** designed”; “we… develop a… **well-structured** codebase”; “our work provides **valuable** practical guidance”. Simply present your work and let the reader make these judgments!
- “data imbalance learning” is not a phrase I have heard. Perhaps “imbalanced learning”?
- “unified, strictly formulated, and comprehensive benchmark”. Unsure what “strictly formulated” means. Could simply say “unified, comprehensive benchmark”
- “we build a… codebase…, which implements over 30 methods… and evaluated on 12… datasets”. It seems that “evaluated on” is the wrong tense; also, what is being evaluated?
- This does not belong in an abstract: “We hope this codebase serves as a comprehensive and reproducible benchmark, encouraging further advancements in long-tailed medical image learning.”
- Often unnecessary inclusion of “the” before concepts: “The deep learning techniques”; “the collected image datasets”; “the long-tailed imbalance”
- “The deep learning techniques have proven effective for most computer vision tasks benefiting from the grown-up dataset scale.” Remove “The”; what does “grown-up dataset scale” mean? “Grown-up” is not the right adjective – be more concrete.
- Refrain from claims like “always result” (line 57) – soften to “usually” or similar
- Confused by this justification: “it is vital to recognize these rare diseases in real-world practice, as they are relatively rare for doctors and may also lack diagnostic capacity.” This reads as “it is vital to recognize rare diseases because they are rare”.
- Line 65: can change “contributions, i.e.,” -> “contributions:”
- Be consistent with capitalization/presentation of terms: “Re-sampling” vs “re-sampling”; “Module improvement” vs. “Module Improvement”; “mnist” vs. “MNIST”; “mixup” vs. “MixUp”; etc.
- Line 82: “we are still curious to explore”. Perhaps just “we aim to explore”?
- “The partition schemes are vita important”. What does “vita” mean?
- The last two paragraphs of the introduction are probably better off being formatted as bulleted or numbered lists. Also, it is unclear why these numbered lists are formatted differently: **1) xxx.** vs. (1) xxx.
- “of class $k$ where $\rho$ denoted as imbalance ratio”. The phrase “denoted as” is awkward + need a comma after $k$
- “a common assumption in long-tailed learning is when the classes are sorted by cardinality in decreasing order” I’m not sure what this means or why this represents an “assumption”. I would just remove this sentence since it does not seem to be used later.
- Line 144: “is a long-tailed version constructed from”. Need to say it is a version “of” something; alternatively, use a word other than “version” like “dataset”
- Inconsistent spacing/use of commas in numbers. “10, 015” -> “10,015”; “3200 fundus images” -> “3,200 fundus images”
- Inconsistent spacing around commas and colons: “training/off-site testing / on-site testin”; “7 : 1: 2”; etc.
- “Liver Tumor Segmentation Benchmark” -> “the Liver Tumor Segmentation Benchmark”
- I realize it is hard to categorize some methods into one bin but GCL loss going in Information Augmentation is interesting, particularly since all other losses fall under re-sampling. It seems to also have module improvement as well.
- “Causal classifier (Tang et al., 2020) resorted to causal inference for keeping the good and removing the bad momentum causal effects in long-tailed learning.” The phrase “resorted to” is strange and has a negative connotation; also, what do “good” and “bad” mean?
- “All these designs are for the fairness and the practicality of the comparison on the benchmark.” Too vague – in what specific way do these support fairness?
- Table 3: Inconsistent “Avg” vs “Avg.” vs “avg”
- Table 4: Consider using a line break occasionally (so one loss function occupies two rows). This would allow you to use a larger font size. Also, be consistent “CrossEntropy” vs “CE”?
- “assessing MixUp based solely on performance is not fair”. Soften to “may not be fair”
- “led to a significant performance decline, e.g,”. Refrain from saying “significant” without statistical significance test + change “e.g,” -> “e.g.,”
- “Use two-stage training as a general paradigm” sounds like a command. Perhaps “Using”?
- Define “SSL” acronym at first use
- “Modify classifier to reduce prediction bias” -> “Classifier modification to reduce prediction bias”
- “In Fig. 2, We visualize” -> “In Fig. 2, we visualize”
- Table 5 indicates the meaning of asterisk, which is never used in the table.
- “models with larger parameters”. The parameters are not “larger” – could say “more parameters” or “a larger parameter count” perhaps.

---

> ### Author Response · Authors · 2024-11-24
> **Reply to Reviewer FbkS (Part 1/N)**
>
> We sincerely appreciate the reviewer’s positive feedback in support of our work. We would also like to thank the reviewer for their valuable efforts and insightful comments, which have greatly contributed to improving our manuscript.
>
> In response to the feedback, we have implemented significant updates, including:
> - Providing an anonymous repository for our codebase: [https://anonymous.4open.science/r/MONICA-153F/README.md](https://anonymous.4open.science/r/MONICA-153F/README.md).
> - Extending support for more advanced backbones, such as ViT, Swin Transformer, ConvNext, and RetFound, alongside detailed experiments to evaluate their impact.
> - Expanding evaluation metrics to include AUROC, AUPRC, and F1 Score, enabling a more comprehensive assessment of model performance.
> - Providing detailed responses and analyses regarding the extended tasks, such as OOD detection.
>
> The detailed responses are outlined below:
>
> > Writing can be improved throughout.
>
> **A:** We sincerely appreciate the detailed feedback and will thoroughly revise the manuscript to address all the highlighted issues and ensure clarity, consistency, and professionalism throughout.
>
> > It is possible that choosing a fixed set of hyperparameters across methods unintentionally advantages certain methods. Ideally, one could argue that each method should be individually tuned on each task;
>
> **A:** Thank you for raising this important point. We acknowledge that using a fixed set of hyperparameters across methods may inadvertently favor certain approaches. While ideally, each method should be individually tuned for each task to maximize its potential, this can introduce inconsistencies and make comparisons less fair, as tuning procedures vary widely across methods and datasets. To strike a balance, we chose to use standardized hyperparameters as a baseline for fair and reproducible comparisons, but we also encourage researchers to explore task-specific tuning using our codebase, which provides the flexibility to adjust hyperparameters for individual methods.
>
> We will also include this point as one of the **limitations** of our work in the discussion section.
>
> > Uncertainty estimates should be provided (e.g., bootstrapped confidence intervals or standard deviations over multiple runs);
>
> **A:** Apologies for the missing details. We ran each methodology on each dataset three times using different random seeds. In the updated version, we will include error bars (e.g., rounded to one decimal place) for greater clarity. Additionally, we plan to create a website to provide more comprehensive results, offering a better overall understanding of this benchmark.
>
> > Multiple performance metrics should be provided (e.g., AUROC).
>
> **A:** In the updated codebase, we expanded the evaluation metrics to include **AUROC, AUPRC, and F1 Score**, providing a more comprehensive assessment of model performance.
>
> >Can the authors provide a summary of practical suggestions for which methods to use in a few sentences near the Conclusion?
>
> **A:** As discussed in the experimental section, we conducted an in-depth analysis from multiple perspectives, including the use and summarization of re-sampling strategies, employing appropriate data augmentation (e.g., MixUp) to enhance the backbone's representation capability, improving classifiers to mitigate distribution bias, and the necessity of two-stage training methods. Through these analyses, we not only derived many significant conclusions but also identified potential improvement directions for each component, providing a broader design space for future researchers to develop new methods. At the same time, we highlighted the importance of the intrinsic characteristics of medical data, such as incorporating prior knowledge as auxiliary information.
>
> The key conclusions are summarized in the following two paragraphs before the Sec. Conclusion:
>
> - **What makes an essential strong baseline?**: We synthesize our findings into actionable insights for building robust baselines in long-tailed medical image classification.
> - **Integration of medical domain prior knowledge**: We emphasize the importance of incorporating domain knowledge (e.g., hierarchical relationships) as a key driver for further improvements in medical imaging tasks.
>
> We have also revised structure and hope this will address the concerns.
>
> #### **4.2.1 Overall Performance**
> - **Overall performance evaluation**
> - **Curse of shot-based group evaluation**
>
> #### **4.2.2 Decoupled Components Discussion**
> - **Effectiveness of re-sampling across SOTAs**
> - **Use two-stage training as a general paradigm**
> - ...
>
> #### **4.2.3 Extended Tasks**
> - **LTMIC improves out-of-distribution detection**
> - **Using imbalanced validation dataset for checkpoint selection**
> -  ...
>
> ####  **4.2.4 Overall Conclusion**
> - **What makes an essential strong baseline?**
> - **Integration of medical domain prior knowledge**
>
> ### **Sec. 5 Conclusion -> Sec. 5 Discussion**

---

> ### Author Response · Authors · 2024-11-24
> **Reply to Reviewer FbkS (Part 2/N)**
>
> > I might suggest including the rank of each method on a given task in all tables.
>
> **A:** Thank you for your suggestion. Initially, we planned to rank the methods; however, as discussed in the **"Curse of Shot-Based Group Evaluation"**, different scenarios prioritize different shot-based groups, making it challenging to claim that methods with higher average accuracy are universally superior. Therefore, we chose to highlight only the best-performing methods in bold and left room for further exploration of new evaluation metrics in future research. Additionally, we plan to incorporate a ranking function into the benchmark website in future updates to enhance usability.
>
> > More details on OOD detection task?
>
> **A:** We advocate that OOD detection is an extension task that will interest researchers working on long-tailed problems, especially considering the challenges that long-tailed issues bring to OOD detection tasks. Mathematically, OOD detection typically relies on the model's predictive confidence $p(y | x)$ to differentiate ID and OOD samples.
>
> OOD (Out-of-Distribution) detection is a task where a model is trained on a known dataset \( D_{in} \) (in-distribution data) and aims to identify whether a given input \( x \) belongs to \( D_{in} \) or an unknown, unseen distribution \( D_{OOD} \) (out-of-distribution data). The core objective can be explained mathematically as follows:
>
> ### Objective of OOD Detection
> **Definition of In-Distribution and Out-of-Distribution:**
>
> In-distribution (ID) data: $D_{in} = \{ (x_i, y_i) \}_{i=1}^N$,
>
> where $x_i \in X_{in}$ and $y_i \in Y_{in}$ are drawn from a distribution $P_{in}(x, y)$.
>
> Out-of-distribution (OOD) data: $D_{OOD} = \{ x_j \}_{j=1}^M$,
>
> where $ x_j \notin X_{in}$ and $ y_j \notin Y_{in} $, sampled from $P_{OOD}(x) $.
>
> **Predictive Confidence:**
>    OOD detection typically relies on the predictive confidence $ p(y | x)$ of the model: $p(y | x) = \frac{\exp(f_y(x))}{\sum_{k \in Y_{in}} \exp(f_k(x))}$,
>
> where $ f_k(x) $ is the logit (pre-activation score) for class $ k $, and $ Y_{in} $ represents the set of known classes.
>
> **Decision Rule:**
>    The task is to define a decision boundary $tau$ such that:
>    $
>    g(x) =
>    \begin{cases}
>    ID, & \text{if } \max_{y \in Y_{in}} p(y | x) \geq \tau, \\
>    OOD, & \text{if } \max_{y \in Y_{in}} p(y | x) < \tau.
>    \end{cases}
>    \$
>
>    Here, $ \tau$ is a threshold, often determined through validation on an auxiliary OOD dataset.
>
> **Evaluation Metrics:**
>    OOD detection is evaluated using metrics such as **AUROC:**
> $
>      AUROC = \int_{0}^{1} TPR(FPR^{-1}(t)) \, dt,
> $
>  where `TPR` and `FPR` are the true positive and false positive rates, respectively.
>
>  In our cases, we have 1000 OOD samples from ImageNet as $D_{OOD}$, where $M$ = 1000, and test set from `OrganAMNIST` as $D_{in}$. All evaluated OOD methods will give a predictive probability of $p(y | x)$.
>
>
> In long-tailed settings, this confidence distribution is skewed due to class imbalance.
>
> For head classes (frequent classes), the model is well-calibrated, producing high confidence:
>
> $
> p(y_{head} | x_{head}) ≈ 1.
> $
>
> For tail classes (rare classes), the model often produces low and uncertain confidence due to insufficient training data:
>
> $
> p(y_{tail} | x_{tail}) ≪ 1.
> $
>
> Consequently, the confidence for tail-class ID samples may overlap significantly with that of OOD samples:
>
> $
> p(y_{tail} | x_{tail}) ≈ p(y_{OOD} | x_{OOD}),
> $
>
> leading to poor separation between ID and OOD samples.
>
> In this context, methods designed for long-tailed problems adjust decision boundaries to account for class imbalances. This prevents the model from overly favoring head classes and ensures that tail classes are not confused with OOD samples. A well-calibrated model trained with long-tailed techniques is more likely to differentiate:
>
> $
> p(y_{tail} | x_{tail}) > p(y_{OOD} | x_{OOD}).
> $
>
> To conclude, Long-tailed problems in real-world scenarios aim to include as many categories as possible (e.g., rare diseases). However, there is still the potential for unknown categories to exist. The goal of OOD detection is to identify these unknown categories, thereby **achieving broader information coverage**. This aligns with the motivation of addressing long-tailed problems.
>
> We will include the definition and provide clearer experimental details regarding this in our supplementary documents.

---

> ### Author Response · Authors · 2024-11-24
> **Reply to Reviewer FbkS (Part 3/N, N=3)**
>
> >More explanations on Fig.4
>
> **A:** Thank you for highlighting these points. We would like to provide further clarification on the purpose of using an imbalanced validation dataset for checkpoint selection and the reasoning behind our conclusions.
>
> This approach addresses a key difference between long-tailed practices in natural images and medical images. In natural image datasets like CIFAR-LT and ImageNet-LT, the original data distribution is balanced, allowing researchers to construct balanced validation sets. This makes evaluation using metrics like loss value, macro accuracy, or group accuracy straightforward and fair. However, in medical imaging, creating a balanced validation set for checkpoint selection is extremely challenging due to the inherent imbalance in the data. Using an imbalanced validation set, while more challenging, better reflects real-world scenarios and aligns with the nature of long-tailed medical datasets.
>
> Additionally, sampling strategies in long-tailed tasks can lead to instability during training. For instance, some methods, e.g, RW, rely on fixed sampling weights determined at the beginning of training, while others adjust sampling dynamically during training, e.g, focal loss, causing fluctuations in performance. Therefore, beyond measuring a method's absolute performance, its stability during training is also a critical factor to consider.
>
> Given the lack of standardized evaluation methods in the community, this perspective ensures that checkpoint selection and performance evaluation are aligned with real-world data distributions, rather than relying solely on artificial or overly idealized validation settings.
>
> Regarding Figure 4, we apologize for the oversight. The labels for the `x` and `y` axes should be `AUROC value`. We will revise the figure. We also note that the conclusion regarding GCL exhibiting lower fluctuations highlights the importance of stability for practical deployment, which we will make more explicit in the revised manuscript.
>
> ### Other small questions
>
> > Avoid editorializing with value judgments: “benchmark is meticulously designed”; “we… develop a… well-structured codebase”; “our work provides valuable practical guidance”. Simply present your work and let the reader make these judgments!
>
> **A:** Thank you for pointing this out. We will remove relevant claims.
>
> > “a common assumption in long-tailed learning is when the classes are sorted by cardinality in decreasing order” I’m not sure what this means or why this represents an “assumption”. I would just remove this sentence since it does not seem to be used later.
>
> **A:** Some datasets were originally assigned labels from 0 to 9 (assuming there are ten classes), but these labels were not sorted by the number of samples in descending order. We would like to note that, for the sake of performance calculation, we have re-ordered the categories in the modified datasets. We provided scripts to present how we build this process in `MONICA/utils/`.
>
> **We hope the above discussion will fully address your concerns about our work.** We look forward to your insightful and constructive responses to further help us improve the quality of our work. Thank you!

---

> ### Comment · Reviewer_FbkS · 2024-11-25
>
> Thank you to the authors for the detailed rebuttal. To clarify one comment, my request for additional evaluation metrics was not meant with respect to the codebase but rather the paper itself: ideally include results with AUROC/balanced accuracy/another appropriate metric in the supplemental materials. I am aware that group-wise accuracy is the primary evaluation metric in the long-tailed learning setting, so I support its use throughout the main text; however, it is entirely possible that a different metric may tell a different story and lead to new interpretations than accuracy alone.
>
> I still do not quite understand the last point regarding sorting labels or what "order" even means in this context. The only impact that the order of labels would have is on how one might index the list of labels to assess the subgroup of interest -- it does not impact the results whatsoever. I think this only causes confusion and represents a trivial implementation detail of performance metric calculation (whether to subset for consecutive elements of a list or not), unless I am misunderstanding something.
>
> Overall, this is a strong submission and I will maintain my original score.

---

> > ### Author Response · Authors · 2024-11-27
> > **Reply to Reviewer FbkS**
> >
> > > It is entirely possible that a different metric may tell a different story and lead to new interpretations than accuracy alone.
> >
> > **A:** Yes! We would add some discussion on the performance of different methods regarding different metrics. For example, when a certain metric is comparable, we can use another metric to evaluate the trade-offs of a method in different scenarios. Specifically, considering the unique challenges of long-tailed problems in low-confidence predictions, we employ various metrics to assess whether the model is well-calibrated [1], [2].
> >
> > > "Oder" for sorting labels.
> >
> > **A:** The order does not affect the results but is relevant to how we preprocess the data and present the quantitative results. We apologize for any confusion caused and will remove redundant descriptions to improve clarity.
> >
> >
> > [1] "On Calibration of Modern Neural Networks" (Guo et al., 2017)
> >
> > [2] "Towards Calibrated Model for Long-Tailed Visual Recognition" (Hong et al., 2021)
> >
> > **We sincerely appreciate your positive recognition and assessment of our work!**

---

### Official Review · Reviewer_gxGu · 2024-11-01

**Soundness:** 3
**Presentation:** 2
**Contribution:** 2
**Rating:** 5
**Confidence:** 5

**Summary:**

This work introduced Medical OpeN-source Long-taIled ClassifiCAtion (MONICA), a comprehensive benchmark for long-tailed medical image classification (LTMIC). It includes a unified, well-structured codebase integrating over 30 methods developed in relevant
fields and 12 long-tailed medical datasets covering 6 medical domains.

**Strengths:**

Long-tailed learning is an extremely challenging problem, this work can serves as a comprehensive and reproducible benchmark, encouraging further advancements in long-tailed medical image learning.

It covers most of the strategies that deal with long-tailed problems, and also include 12 datasets from different application domains.

**Weaknesses:**

This work doesn't introduce any new datasets or methods. It is a collection of datasets (multi class or multi label) that are already publicly available without justifications as they are many other such kind of long tail datasets available. Also, they have changed some of the original datasets, it would not be useful if they don't share the modified datasets.

They only tried ResNet for the tasks, would be nicer to make comparisons with other models. Also the discussions on SSL models seem not supported by any data.

**Questions:**

Some of the datasets have been changes in terms of distributions, would you share the modified datasets or the code on making the changes?

What are the performance of the SSL models?

There are some typos such as quotation markers etc, please correct.

**Details Of Ethics Concerns:**

All public datasets, no concerns.

---

> ### Author Response · Authors · 2024-11-24
> **Reply to Reviewer gxGu (Part 1/N)**
>
> We thank the reviewer for the positive opinions and helpful suggestions.
>
> We have conducted extensive updates and additional experiments based on the reviewers' valuable feedback. Specifically,
>
> - We added an anonymous repository for our codebase: https://anonymous.4open.science/r/MONICA-153F/README.md
> - We released the download link, train/val/test split and preprocessing codes for all the datasets.
> - We added support for more advanced backbones, including ViT, Swin Transformer, ConvNext, and RetFound, and performed detailed experiments to evaluate their impact.
> - We added the results for SSL methods.
>
> The detailed responses are outlined below:
>
> ### Main Questions:
> > This work doesn't introduce any new datasets or methods. It is a collection of datasets (multi class or multi label) that are already publicly available without justifications as they are many other such kind of long tail datasets available. Also, they have changed some of the original datasets, it would not be useful if they don't share the modified datasets.
>
> **A:** We have released the download link, train/val/test split and preprocessing codes for all the datasets. We also provided a guide for the users to build their own customized datasets for training. We hope this can address your questions.
>
>
>
> >They only tried ResNet for the tasks, would be nicer to make comparisons with other models.
>
> **A:** Unless explicitly specified in the original paper that a particular backbone must be used (e.g., **RSG uses ResNext**), we choose ResNet as our backbone for the following reasons: the methods supported by our benchmark predominantly use ResNet as the backbone in their original papers. This is especially important for methods that require modifications to the network structure (e.g., the **ResNet** in **SADE**). Attempting to modify ViT in the same way as ResNet to achieve a similar motivation could introduce discrepancies, potentially leading to **unfair evaluations**.
>
> However, we appreciate the suggestions from the reviewer to add more backbones with different architectures. Our updated codebase now supports more advanced backbones such as ViT, Swin Transformer, ConvNext, as well as relevant foundation models like RetFound for ophthalmology. We have provided corresponding alerts in the codebase for those fixed backbones proposed by the original paper. We completed training for **ViT, Swin Transformer, and ConvNext** on the ISIC dataset. Here, we present partial results, and the complete results will be included in the appendix.

---

> > ### Author Response · Authors · 2024-11-24
> > **Reply to Reviewer gxGu (Part 2/N)**
> >
> > **Table 1. ISIC2019-LT (r=100)**
> > |     Methods     | ResNet50 | ViT-Base | SwinTrans-Base | ConvNext-Base |
> > |:---------------:|:--------:|:--------:|:--------------:|:-------------:|
> > |       ERM       |  59.33   |  53.17   |     68.27      |     74.05     |
> > |        RS       |  60.06   |  59.55   |     72.41      |     77.14     |
> > |        RW       |  59.00   |  38.09   |     31.27      |     63.32     |
> > |      Focal      |  57.28   |  35.86   |     71.05      |     63.90     |
> > |     CB-Focal    |  60.44   |  40.93   |     52.84      |     52.43     |
> > |     LADELoss    |  58.17   |  53.17   |     73.71      |     75.80     |
> > |       LDAM      |  58.61   |  32.79   |     51.43      |     70.68     |
> > | BalancedSoftmax |  59.28   |  56.51   |     60.98      |     69.32     |
> > |      VSLoss     |  56.67   |  56.37   |     73.71      |     56.18     |
> > |      MixUp      |  54.78   |  27.95   |     73.12      |     63.65     |
> > |      MiSLAS     |  55.83   |  54.43   |     64.98      |     67.82     |
> > |       GCL       |  64.06   |  57.27   |     70.07      |     72.73     |
> > |       cRT       |  63.28   |  54.41   |     59.00      |     76.10     |
> > |       LWS       |  56.83   |  59.84   |     67.98      |     77.37     |
> > |       KNN       |  61.44   |  56.78   |     65.06      |     66.39     |
> > |       LAP       |  61.22   |  44.85   |     73.38      |     73.95     |
> > |   De-Confound   |  59.67   |  55.62   |     72.18      |     57.91     |
> > |     DisAlign    |  64.44   |  56.17   |     72.13      |     80.64     |
> >
> > **Table 2. ISIC2019-LT (r=200)**
> > |     Methods     | ResNet50 | ViT-Base | SwinTrans-Base | ConvNext-Base |
> > |:---------------:|:--------:|:--------:|:--------------:|:-------------:|
> > |       ERM       |  54.06   |  28.70   |     55.28      |     65.02     |
> > |        RS       |  58.33   |  50.91   |     52.97      |     71.92     |
> > |        RW       |  56.50   |  32.74   |     19.02      |     43.21     |
> > |      Focal      |  54.50   |  29.28   |     63.97      |     56.86     |
> > |     CB-Focal    |  58.50   |  46.28   |     53.70      |     58.29     |
> > |     LADELoss    |  51.67   |  48.90   |     66.86      |     63.28     |
> > |       LDAM      |  54.94   |  55.25   |     49.61      |     55.06     |
> > | BalancedSoftmax |  61.89   |  49.85   |     65.24      |     67.89     |
> > |      VSLoss     |  53.39   |  28.70   |     66.86      |     68.52     |
> > |      MixUp      |  50.33   |  51.97   |     67.39      |     70.89     |
> > |      MiSLAS     |  56.50   |  56.92   |     51.00      |     70.40     |
> > |       GCL       |  64.00   |  53.81   |     68.09      |     68.99     |
> > |       cRT       |  60.22   |  60.55   |     72.20      |     67.67     |
> > |       LWS       |  54.50   |  51.83   |     57.11      |     68.39     |
> > |       KNN       |  58.33   |  56.28   |     62.72      |     61.50     |
> > |       LAP       |  58.39   |  54.98   |     66.86      |     63.26     |
> > |   De-Confound   |  53.17   |  55.25   |     64.98      |     57.36     |
> > |     DisAlign    |  63.33   |  48.15   |     64.80      |     57.48     |
> >
> > **Table 3. ISIC2019-LT (r=500)**
> > |     Methods     | ResNet50 | ViT-Base | SwinTrans-Base | ConvNext-Base |
> > |:---------------:|:--------:|:--------:|:--------------:|:-------------:|
> > |       ERM       |  45.78   |  47.16   |     48.98      |     61.35     |
> > |        RS       |  47.39   |  42.36   |     63.29      |     60.05     |
> > |        RW       |  44.78   |  27.48   |     16.89      |     47.84     |
> > |      Focal      |  46.67   |  26.06   |     56.27      |     63.34     |
> > |     CB-Focal    |  48.39   |  35.27   |     17.33      |     42.82     |
> > |     LADELoss    |  45.17   |  47.16   |     41.12      |     56.33     |
> > |       LDAM      |  45.72   |  50.15   |     50.41      |     42.18     |
> > | BalancedSoftmax |  51.11   |  35.35   |     49.32      |     55.70     |
> > |      VSLoss     |  45.89   |  47.16   |     41.12      |     33.87     |
> > |      MixUp      |  44.33   |  45.32   |     30.44      |     40.46     |
> > |      MiSLAS     |  48.61   |  50.54   |     30.62      |     61.52     |
> > |       GCL       |  54.83   |  50.75   |     49.02      |     52.68     |
> > |       cRT       |  46.61   |  48.27   |     50.31      |     60.79     |
> > |       LWS       |  47.00   |  49.50   |     47.67      |     59.38     |
> > |       KNN       |  49.22   |  44.44   |     55.72      |     56.06     |
> > |       LAP       |  49.56   |  42.53   |     57.02      |     59.72     |
> > |   De-Confound   |  42.50   |  48.72   |     53.26      |     61.61     |
> > |     DisAlign    |  51.72   |  44.37   |     46.39      |     49.01     |

---

> > > ### Author Response · Authors · 2024-11-24
> > > **Reply to Reviewer gxGu (Part 3/N)**
> > >
> > > **The Key Observations:**
> > >
> > > - **ViT-Base struggles** in most settings and methodologies compared to ConvNext and SwinTrans. This is due to that transformer relies on **more datasets** for training.
> > > - But the transformer structure is not necessarily worse than CNN. Swin Transformer base achieves higher performance than the ResNet50 in terms of most settings and methods.
> > > - **ConvNext-Base** consistently outperforms other backbones (ResNet50, ViT-Base, and SwinTrans-Base) across many methods, especially for more complex scenarios. This suggests that modern architectures like ConvNext-Base are more robust for handling diverse long-tailed datasets.
> > > - Methods involving **model modifications** often experience **significant performance degradation** or **show less noticeable improvements** when applied to **new backbones** (i.e., backbones different from those used in the original papers). For example, when adapting `GCL` (which performs well on most datasets using `ResNet50`), modifying `ResNet50` typically involves directly adjusting the `model.fc` layer. However, when adapting `GCL` to `Swin Transformer`, there are two possible approaches: either modify the entire `classifier head` or adjust only the `fc` layer within the `classifier head`. Here, we present the results of the better-performing approach for each case, but no unified conclusion supports whether the former or the latter is superior. Therefore, for methods that involve network modifications, it is still recommended to **use the network architecture specified in the original paper**.
> > >
> > > Based on the experimental results with the aforementioned backbones, it is evident that stronger backbones can indeed help learn better features. However, due to differences in network architecture (**CNN-based, transformer-based**) and the number of parameters, it is challenging to directly compare their differences horizontally. To address this, we added a new set of results to specifically compare the performance differences between **ResNet50** and **ResNet101** across methods.
> > >
> > >
> > >
> > > **Table 4. ISIC2019-LT (r=100)**
> > > |     Methods     | ResNet50 | ResNet101 |
> > > |:---------------:|:--------:|:-----------:|
> > > |       ERM       |  59.33   | 59.25     |
> > > |        RS       |  60.06   | 63.94     |
> > > |        RW       |  59.00   | 55.18     |
> > > |      Focal      |  57.28   | 60.13     |
> > > |     CB-Focal    |  60.44   | 59.14     |
> > > |     LADELoss    |  58.17   | 62.59     |
> > > |       LDAM      |  58.61   | 59.93     |
> > > | BalancedSoftmax |  59.28   | 63.08     |
> > > |      VSLoss     |  56.67   | 44.51     |
> > > |      MixUp      |  54.78   | 65.35     |
> > > |      MiSLAS     |  55.83   | 59.06     |
> > > |       GCL       |  64.06   | 72.22     |
> > > |       cRT       |  63.28   | 56.60     |
> > > |       LWS       |  56.83   | 36.69     |
> > > |       KNN       |  61.44   | 63.19     |
> > > |       LAP       |  61.22   | 60.88     |
> > > |   De-Confound   |  59.67   | 60.21     |
> > > |     DisAlign    |  64.44   | 65.27     |
> > >
> > > **Table 5. ISIC2019-LT (r=200)**
> > > |     Methods     | ResNet50 | ResNet101 |
> > > |:---------------:|:--------:|:-----------:|
> > > |       ERM       |  54.06   | 56.16     |
> > > |        RS       |  58.33   | 58.58     |
> > > |        RW       |  56.50   | 47.16     |
> > > |      Focal      |  54.50   | 50.29     |
> > > |     CB-Focal    |  58.50   | 53.37     |
> > > |     LADELoss    |  51.67   | 39.30     |
> > > |       LDAM      |  54.94   | 55.95     |
> > > | BalancedSoftmax |  61.89   | 58.76     |
> > > |      VSLoss     |  53.39   | 43.53     |
> > > |      MixUp      |  50.33   | 61.46     |
> > > |      MiSLAS     |  56.50   | 54.42     |
> > > |       GCL       |  64.00   | 63.86     |
> > > |       cRT       |  60.22   | 59.96     |
> > > |       LWS       |  54.50   | 44.55     |
> > > |       KNN       |  58.33   | 59.11     |
> > > |       LAP       |  58.39   | 60.45     |
> > > |   De-Confound   |  53.17   | 55.88     |
> > > |     DisAlign    |  63.33   | 60.26     |
> > >
> > > **Table 6. ISIC2019-LT (r=500)**
> > > |     Methods     | ResNet50 | ResNet101 |
> > > |:---------------:|:--------:|:-----------:|
> > > |       ERM       |  45.78   |   52.71   |
> > > |        RS       |  47.39   |   45.90   |
> > > |        RW       |  44.78   |   37.24   |
> > > |      Focal      |  46.67   |   33.18   |
> > > |     CB-Focal    |  48.39   |   37.47   |
> > > |     LADELoss    |  45.17   |   23.72   |
> > > |       LDAM      |  45.72   |   49.82   |
> > > | BalancedSoftmax |  51.11   |   50.54   |
> > > |      VSLoss     |  45.89   |   32.99   |
> > > |      MixUp      |  44.33   |   57.21   |
> > > |      MiSLAS     |  48.61   |   47.36   |
> > > |       GCL       |  54.83   |   59.86   |
> > > |       cRT       |  46.61   |   55.52   |
> > > |       LWS       |  47.00   |   38.93   |
> > > |       KNN       |  49.22   |   49.11   |
> > > |       LAP       |  49.56   |   51.16   |
> > > |   De-Confound   |  42.50   |   50.29   |
> > > |     DisAlign    |  51.72   |   42.99   |

---

> ### Author Response · Authors · 2024-11-24
> **Reply to Reviewer gxGu (Part 4/4, N=4)**
>
> The key observations:
> - Overall, increasing the number of parameters does improve performance for most methods, particularly for MixUp. This is likely because MixUp, as a data augmentation and regularization technique, benefits from greater model capacity, allowing it to learn more robust feature representations.
>
> - In terms of efficiency, as the level of imbalance increases, simply increasing model capacity shows diminishing returns, indicating the need to explore other strategies to enhance performance. Additionally, under the `Pareto distribution sampling setting`, the increase in imbalance significantly reduces the number of training samples, which may lead to **overfitting** in larger models. Therefore, **the number of model parameters** should be appropriately aligned with **the amount of training data** to achieve optimal performance.
>
> > Also the discussions on SSL models seem not supported by any data.
>
> **A:** We apologize for the missing experimental results on SSL methods.
>
> The following are the SSL results. Our codebase does not support all SSL methods directly; instead, the methods mentioned in other papers are pre-trained using their official code and then loaded into MONICA as initialized weights for further training.
>
> **Table 7. Experimental results on ISIC2019-LT (r=100) trained from different self-supervised methods.**
> | SSL Methods/Weights |     Source Data     | ISIC2019-LT (r=100) |
> |:-------------------:|:-------------------:|:-------------------:|
> | Supervised ImageNet |       ImageNet      |        59.33        |
> |         BYOL        | ISIC2019-LT (r=100) |         16.70        |
> |        MOCOv2       | ISIC2019-LT (r=100) |         25.60        |
> |        SimCLR       | ISIC2019-LT (r=100) |         11.10        |
> |         KCL         | ISIC2019-LT (r=100) |        50.83        |
>
> As we discussed in our manuscript, we conclude this for several reasons: (1) The amount of medical imaging data in this study is **limited**, especially for a LTMIC setting, while SSL often relies on large quantities of unlabeled data. (2) SSL typically depends on **specific hyper-parameter designs/tuning** such as data augmentations, which are crucial to the second-stage fine-tuning. However, the lack of guidance in these aspects diminishes the possibility of achieving optimal practice.
>
> In the future work, we will explore some SSL methods designed for the **specific medical domain**, e.g., Li et al. for ophthalmology [1].
>
> [1] Li, Xiaomeng, et al. "Self-supervised feature learning via exploiting multi-modal data for retinal disease diagnosis." IEEE Transactions on Medical Imaging 39.12 (2020): 4023-4033.
>
> >There are some typos such as quotation markers etc, please correct.
>
> A: Thank you for pointing this out. We will thoroughly review the manuscript to address any typos and ensure clarity.
>
> **We hope the above discussion will fully address your concerns about our work, and we would really appreciate it if you could be generous in raising your score.** We look forward to your insightful and constructive responses to further help us improve the quality of our work. Thank you!

---

> ### Author Response · Authors · 2024-11-27
> **Inquiry about further concerns/questions**
>
> Dear reviewer gxGu,
>
> We would like to know if our response has addressed your concerns and questions. If you have any further concerns or suggestions for the paper or our rebuttal, please let us know. We would be happy to engage in further discussion and manuscript improvement.
>
> Thank you again for the time and effort you dedicated to reviewing this work.
>
> Sincerely,
>
> The Authors

---

> > ### Author Response · Authors · 2024-11-30
> > **Gentle Reminder of the Discussion Deadline**
> >
> > Dear Reviewer gxGu,
> >
> > Thank you once again for your time! We understand that you have a busy schedule, and we kindly remind you that the discussion deadline is approaching. If you have any suggestions or feedback on how we can improve our manuscript, we would greatly appreciate your input. We eagerly await your response.
> >
> > Sincerely,
> >
> > The Authors

---

> > > ### Author Response · Authors · 2024-12-01
> > > **Gentle Reminder of the Discussion Deadline - Final Day**
> > >
> > > Dear Reviewer gxGu,
> > >
> > > We understand that the timing of this discussion period may not align perfectly with your schedule.
> > >
> > > As the discussion deadline is approaching, could you kindly let us know if your concerns have been adequately addressed? If not, we are happy to provide further clarification. If you feel your concerns have been resolved, we would greatly appreciate it if you could reconsider your review score.
> > >
> > >
> > > Sincerely,
> > >
> > > The Authors

---

### Official Review · Reviewer_3RRv · 2024-11-03

**Soundness:** 2
**Presentation:** 2
**Contribution:** 2
**Rating:** 5
**Confidence:** 4

**Summary:**

This paper presents a unified benchmark for long-tailed learning in the medical domain by integrating several existing datasets and implementing a complete pipeline from data loading to model training and evaluation. The authors claim that the benchmark supports over 30 methods for comparison and provides an analysis of their performances.

- Update after the discussion phase:

Thank you for the detailed responses! I've raised my score. While my concerns are not entirely resolved, I believe with careful revisions, its future version has the potential to be accepted.

**Strengths:**

The paper attempts to provide a comprehensive benchmark for long-tailed medical image classification. The idea of integrating multiple existing methods and datasets into a unified platform could potentially be useful for researchers who want to compare various methodologies under a standardized framework.

**Weaknesses:**

1. Motivation. The paper lacks sufficient justification for evaluating long-tailed problems specifically in medical imaging tasks. While the authors mention some motivations at the beginning, these arguments are not convincing. Is there a fundamental difference between long-tailed problems in medical imaging and those in conventional tasks? Would this difference necessitate different methodologies? Even if the data modalities and evaluation methods are distinct (e.g., balanced vs. imbalanced test sets), would this lead to fundamentally different approaches? The paper analyzes multiple methods based on this premise but fails to provide insightful conclusions, which further deepens my skepticism about the motivation.

2. Dataset Contribution. Although the paper claims to use 12 datasets, 7 of these come from MedMNIST, and several of them are derived from previous work. This reduces the originality of the dataset contribution. Furthermore, the split between multi-class and multi-label datasets is 9/3, respectively. It is worth noting that many existing studies have already utilized MedMNIST for long-tailed learning (https://scholar.google.com/scholar?cites=11226954386823169312&scipsc=1&q=long+tail). Given that 7 out of the 12 datasets in this paper are from MedMNIST, why should users choose MONICA over MedMNIST, which already has extensive use and coverage in the medical imaging field? Additionally, the experimental methods used for multi-class and multi-label datasets are almost entirely different, and the analysis of multi-label results is limited to a single vague statement that multi-label classification is more challenging. This gives the impression that multi-label datasets were included just for the sake of completeness, rather than being a key focus.

3. Code Contribution. The code is not provided in the appendix, nor is there an anonymous GitHub link, which means the authors' claims about the code cannot be verified. By comparison, the NeurIPS D&B track (single-blind review) usually includes dataset or code links, along with information about author affiliations, licenses, and ethics. Although such links may be added after acceptance, this suggests that such work may not be well-suited for ICLR's double-blind review process.

    Additionally, the description of the code structure in Section 3.1 is not particularly informative. The modular design described is basic and lacks novel insights. A more impactful modular design, like in mmdetection, which breaks down components into backbone, neck, and bbox head, would have been more meaningful. As it stands, the description feels unnecessary.

4. Experimental Analysis Lacks of Insights. Comments below:

    - Despite using multiple datasets, the authors only provide a generic / systematic comparison of the methods without analyzing differences across domains. For example, there is no discussion about which methods are better suited for dermatology versus ophthalmology. Almost all discussion is very general, without any specific insights related to medical applications. This diminishes the value of using 12 datasets, as the conclusions drawn are not substantially different from what could be obtained from a single dataset.

    - The analysis in Section 4.2 is poorly organized. There is no clear structure, with the discussion jumping from evaluation metrics (e.g., "Curse of shot-based group evaluation") to re-sampling methods, MixUp, two-stage training, and even self-supervised learning in a seemingly random fashion. Many claims are also not supported by data. The overall takeaway from the experimental section is unclear, and I did not gain any insights on how to design better models.

    - In Section 4.1, there is inconsistency in the training strategies used: some methods use a unified training strategy, while others use the one specified in the original paper (e.g., SAM, Line 306), with no explanation for this discrepancy.

    - There are issues with the tables, such as Table 2, where it is unclear what methods like ERM, cRT, and LWS represent, as they are not referenced properly. Additionally, Section 3.2.3 does not fully align with the table.

    - The categorization of methods is confusing. The authors categorize methods into three types—class re-sampling, information augmentation, and module improvement—but later mention that re-sampling and MixUp are used in many methods, making the classification in Tables 2/3 somewhat meaningless.

    - The discussion on self-supervised learning (Line 398) appears out of place, as it is not introduced earlier.

    - Similarly, the mention of OOD detection (Line 421) is abrupt and lacks context.

    - The section on using an imbalanced validation dataset for checkpoint selection is unclear about its purpose. The conclusion seems to be that GCL exhibits lower fluctuations, but the reasoning and implications are not well explained. Additionally, Figure 4 lacks labels for the x and y axes, making interpretation difficult.

    - Line 475 suddenly states that multi-label classification is more challenging without providing adequate context or analysis.

    - Line 504 claims that "the most advanced long-tailed learning methods no longer focus on improving a single strategy," but this claim is not well-supported by the preceding analysis.

**Questions:**

See weakness sections. Some more questions below:

Could you elaborate on why the results of self-supervised learning and OOD detection are relevant in this paper? They seem out of place given the main focus on long-tailed classification.

Why did the authors not include a domain-specific analysis (e.g., which methods work better for certain medical fields)? It seems like an important missed opportunity.

---

> ### Author Response · Authors · 2024-11-24
> **Reply to Reviewer 3RRv (Part 1/N)**
>
> We appreciate the reviewer’s constructive comments and valuable suggestions.
>
> In response to the feedback, we have implemented significant updates and conducted additional experiments, including:
>
> - Providing an anonymous repository for our codebase: [https://anonymous.4open.science/r/MONICA-153F/README.md](https://anonymous.4open.science/r/MONICA-153F/README.md).
> - Extending support for more advanced backbones, such as ViT, Swin Transformer, ConvNext, and RetFound, alongside detailed experiments to evaluate their impact.
> - Expanding evaluation metrics to include AUROC, AUPRC, and F1 Score, enabling a more comprehensive assessment of model performance.
> - Conducting additional ablation studies to analyze variations in input size, backbone comparisons, and augmentation techniques like RandAug and AugMix.
> - Providing detailed responses and analyses regarding the extended tasks, including OOD detection and self-supervised learning.
>
> We believe these updates address the concerns raised and significantly enhance the clarity and depth of our study.
>
> The detailed responses are outlined below:
>
> > Motivation. The paper lacks sufficient justification for evaluating long-tailed problems specifically in medical imaging tasks...
>
> **A:** Thank you for highlighting this concern. We appreciate the opportunity to clarify the motivation and address the fundamental differences between long-tailed problems in medical imaging and classical long-tailed learning tasks for natural images.
>
> ### Medical Background
> - **Intrinsic Nature of Medical Imaging Data:** Unlike natural image datasets, long-tailed distributions in medical imaging are a direct result of real-world prevalence rates (In this work, we consider an extreme challenging scenario with higher imbalance ratio under Pareto distribution). Rare diseases naturally appear less frequently, but their accurate diagnosis is critical in clinical settings due to their high-risk nature. This makes the long-tailed challenge in medical imaging inherently different, as it directly impacts patient outcomes and decision-making in healthcare.
>
> - **Importance of Rare Classes**: In medical imaging tasks, rare classes often correspond to severe or life-threatening conditions. Misclassifying these rare cases can have dire consequences, unlike conventional tasks where tail classes may not carry the same level of significance. This necessitates methods that can effectively handle rare but critical samples, beyond merely improving overall performance. For example, in clinical scenarios, many use cases may prioritize sensitivity, whereas screening scenarios often focus on a balance between sensitivity and specificity. To address this, our updated codebase includes additional metrics such as AUROC, AUPRC, and F1 Score.
> - **Integration of Domain Knowledge**: Medical imaging tasks often rely on incorporating hierarchical relationships among diseases (e.g., different stages of a disease) or prior clinical knowledge, which is rarely considered in conventional tasks. This additional layer of complexity requires adapted methodologies that leverage domain-specific insights, further distinguishing medical imaging from other domains.
>
>
> ### Benchmark and Codebase
> -**Lack of Standardization in Dataset Settings**: There is no unified standard for defining the degree of imbalance between head and tail classes in long-tailed datasets (e.g., imbalance ratio). Some studies use artificially generated long-tailed distributions (e.g., Pareto distribution), while others rely on real-world data distributions, making it difficult to directly compare results across studies. Even with the same dataset, differences in preprocessing methods (e.g., data augmentation, cropping, resolution adjustments) can lead to significant performance differences. The lack of standardized preprocessing protocols impacts the reproducibility of results.
>
> -**Evaluation Discrepancy**: Conventional long-tailed tasks often use balanced test sets to evaluate models, focusing on overall accuracy or head-to-tail trade-offs. In contrast, medical imaging typically evaluates models on naturally imbalanced test sets to simulate real-world distributions. This difference shifts the emphasis to metrics like sensitivity, AUROC, and AUPRC, making certain methodologies unsuitable or less effective in the medical context.
>
> -**Empirical Evidence**: Our comprehensive analysis across multiple datasets and methods highlights that existing long-tailed learning approaches do not always generalize well to medical imaging tasks. For example, methods like GCL or MixUp may show inconsistent results across datasets with varying levels of imbalance or complexity. These observations underline the need for tailored approaches specific to the medical domain.
>
> We will incorporate these clarifications into the revised manuscript to strengthen the motivation and provide additional insights into the distinct challenges and requirements of long-tailed medical imaging tasks.

---

> > ### Author Response · Authors · 2024-11-24
> > **Reply to Reviewer 3RRv (Part 2/N)**
> >
> > >Failed to provide insightful conclusions.
> > >The modular design described is basic and lacks novel insights.
> >
> > **A:** As discussed in the experimental section, we conducted an in-depth analysis from multiple perspectives, including the use and summarization of **re-sampling strategies**, employing appropriate data augmentation (e.g., MixUp) to enhance the **backbone's representation capability**, improving **classifiers** to mitigate distribution bias, and the necessity of **two-stage training methods**. Through these analyses, we not only derived many significant conclusions but also identified **potential improvement directions** for each component, providing a broader design space for future researchers to develop new methods.
> >
> > Furthermore, we would like to emphasize that the codebase we provide enables the design, improvements and potential combination of modules, offering greater flexibility. The additional experiments we conducted with **different backbones and data augmentation strategies** also release new insights.
> >
> > Additionally, we explored the extension tasks on key issues relevant to this field, such as **OOD detection**. At the same time, we highlighted the importance of the intrinsic characteristics of medical data, such as incorporating prior knowledge as auxiliary information—for example, hierarchical information[1], which serves as an excellent case in point.
> >
> >
> >
> > >Dataset Contribution.
> >
> > - **Choice of MedMNIST:** While it is true that 7 out of the 12 datasets in MONICA are derived from MedMNIST, this decision was intentional. MedMNIST serves as a widely recognized resource for medical image analysis and allows users to reproduce results with **minimal effort in data preprocessing**. Medical image datasets often originate from multiple institutions or competitions, and their access can be limited due to restrictions or expired links, making reproducibility challenging. By leveraging MedMNIST, we ensure accessibility and ease of use for researchers, similar to how ImageNet is used in natural image tasks despite being extensively studied. However, unlike MedMNIST, which lacks a unified framework for long-tailed research, MONICA provides a systematic benchmark that integrates data preprocessing, methodology, and evaluation metrics, offering a much-needed structure for long-tailed learning in medical imaging.
> > - **Beyond MedMNIST:** To address the diversity of medical imaging tasks, we intentionally included datasets outside of MedMNIST to represent various medical domains. These additional datasets help capture unique challenges in fields such as **pathology, ophthalmology, and dermatology**. Our goal is to balance ease of access with a meaningful representation of real-world medical classification problems, ensuring that the benchmark remains both practical and comprehensive.
> > - **Focus on Practicality and Customization:** One of MONICA’s key contributions lies in its codebase, which enables users to test methods on their own datasets with ease. Many medical datasets are **private and institution-specific**, making it essential for researchers to adapt strategies to their own data. MONICA provides tools and flexible configurations to bridge this gap, focusing not just on the included datasets but on enabling researchers to explore the effects of different methods and strategies on their proprietary data.
> > - **Why Choose MONICA Over MedMNIST?** MedMNIST primarily provides datasets but lacks a standardized framework for long-tailed learning. MONICA goes beyond by: **(1)** Systematically constructing benchmarks for long-tailed medical image classification, including dataset splits, evaluation metrics, and method comparisons. **(2)** Providing unified implementations of long-tailed learning methods, reducing the complexity of integrating and testing approaches. Offering diverse datasets to represent challenges across different medical fields, such as multi-class vs. multi-label problems.
> > **(3)** Supporting extensibility for users to incorporate their own datasets and customize experiments.

---

> > > ### Author Response · Authors · 2024-11-24
> > > **Reply to Reviewer 3RRv (Part 3/N)**
> > >
> > > > Multi-Class vs. Multi-Label Analysis
> > > > Line 475 suddenly states that multi-label classification is more challenging without providing adequate context or analysis.
> > >
> > > **A:** We acknowledge the limited analysis of multi-label datasets in the current version of the paper. Multi-label tasks indeed present unique challenges, such as label co-occurrence and imbalanced joint distributions, which differ significantly from multi-class classification.
> > >
> > > #### Difference:
> > > - **Activation Function:** In general multi-class classification, each input is assigned to one and only one class. This leads to the use of the softmax function, which ensures that the probabilities across all classes sum to 1: $p(y = i | x) = exp(z_i) / Σ_{j=1}^C exp(z_j),$ In general multi-label classification, each input can belong to multiple classes simultaneously. Hence, the sigmoid function is applied independently to each class: $p(y_i = 1 | x) = 1 / (1 + exp(-z_i)), ∀ i ∈ {1, ..., C}.$  where z$_i $ is the logit (pre-activation output) for class `i`, and `C` is the total number of classes. This difference results in a fundamental divergence in some designs such as loss function: most methods for multiclass tasks aim to improve the **confidence for tail classes**, whereas methods for multilabel tasks focus on enhancing the **recognition of negative samples** corresponding to the specific evaluated categories [2]. Therefore, their objectives are inherently different.
> > > - **Threshold Determination in Multi-label Classification:** In multi-class classification, the prediction is straightforward: $ŷ = argmax_i p(y = i | x).
> > > $. However, in multi-label classification, a threshold must be determined for each class to decide whether it is active $y_{i}=1$ or inactive $y_{i}=0$. Hereby bring the question that Should all classes share the same threshold
> > > or should each class have a unique threshold? Tail classes often require lower thresholds to account for their rarity.
> > > - **Co-occurrence Challenges in Multi-label Classification:** In multi-label classification, label co-occurrence poses significant challenges. For example, in medical imaging, certain diseases (e.g., "diabetic macular edema") frequently co-occur with others (e.g., "diabetic retinopathy"). The joint probability of multiple labels in a multi-label setting must account for these dependencies: $p(y | x) = Π_{i=1}^C p(y_i | x, y_{-i}),$ where $y_{-i}$ represents all labels except $y_i$. Naive re-sampling can disrupt these dependencies, creating unrealistic label combinations and degrading model performance, as well as new relative imbalance.
> > >
> > > **Table1. Key Differences Between Multi-Class and Multi-Label Settings in Long-Tailed Learning**
> > >
> > > | Aspect                        | Multi-class                        | Multi-label                                   |
> > > |-------------------------------|-------------------------------------|----------------------------------------------|
> > > | **Activation Function**       | Softmax (sum-to-one constraint)    | Sigmoid (independent probabilities)          |
> > > | **Competition Among Classes** | Tail classes compete with head ones| Each class is independent but may co-occur   |
> > > | **Sampling Challenges**       | Focus on class frequency           | Must account for label co-occurrence         |
> > > | **Prediction**                | Argmax of softmax outputs          | Thresholding sigmoid outputs                 |
> > > | **Threshold Determination**   | Not required                       | Critical for balancing head and tail classes |
> > >
> > >
> > > In long-tail research on natural images, there has been little effort to study multi-label and multi-class tasks together. However, in medical problems, multi-label tasks are often unavoidable. Therefore, this paper aims to provide some insightful introductions to this setting, along with relevant codes and experimental results, to supplement this area rather than simply adding a few datasets.
> > >
> > >
> > > In the revised version (mainly on supplementary files), we will expand on the challenges specific to multi-label classification.
> > >
> > > > Code Contribution. The code is not provided in the appendix, nor is there an anonymous GitHub link, which means the authors' claims about the code cannot be verified.
> > >
> > > **A:** Thank you for pointing this out. We added an anonymous repository for our codebase: https://anonymous.4open.science/r/MONICA-153F/README.md. We added support for more advanced backbones, including ViT, Swin Transformer, ConvNext, and RetFound, and performed detailed experiments to evaluate their impact. Furthermore, we expanded the evaluation metrics to include AUROC, AUPRC, and F1 Score, providing a more comprehensive assessment of model performance.

---

> ### Author Response · Authors · 2024-11-24
> **Reply to Reviewer 3RRv (Part 4/N)**
>
> > Lack of insights on different datasets.
> > Why did the authors not include a domain-specific analysis?
>
> **A:** In addition to the earlier discussion on overall performance, we also tested the impact of different data augmentation methods on the performance of various datasets. The current version of codebase supports RandAug and we plan to support some other augmentation codebase such as **OpenMixup**[3] in the future update.
>
> **Table 2. Results with Different Augmentation Techniques**
> |                   | Weak (Resize) | Crop (Strong) | RandAug | AugMix + Crop |
> |:-----------------:|:-------------:|:-------------:|:-------:|:-------------:|
> | ISIC-2019 (r=100) |     58.72     |     64.06     |  65.89  |     66.34     |
> |       KVASIR      |     86.07     |     85.14     |  87.88  |     85.32     |
> |    OrganAMNIST    |     79.09     |     78.21     |  80.13  |     79.06     |
>
> **The key observations**:
>
> - Using more **complex data augmentation** methods like RandAug and AugMix can further improve performance. For datasets with diverse image patterns, such as dermoscopy images (e.g., with color variations), the performance gains from applying complex augmentations are often greater compared to others such as radiology datasets.
> - Cropping is a common method in data augmentation, but it may not be suitable for all medical datasets, as medical data often relies on **specific regions of lesions** for classification or diagnosis, such as in cases of diabetic retinopathy grading.
> - Skin lesions are often characterized by certain specific local features, such as the shape of the lesion, irregular edges, and color distribution. By applying **cropping**, the model can focus more on the detailed features of the lesion area, reducing background interference and thereby improving classification or detection performance.
>
> Finally, we would like to emphasize that incorporating prior knowledge in medical imaging can be very helpful in addressing long-tail problems. We acknowledge that one **limitation** of this paper is the inability to incorporate extensive medical knowledge for different datasets. We will address this point in the revised version of the paper.

---

> ### Author Response · Authors · 2024-11-24
> **Reply to Reviewer 3RRv (Part 5/N)**
>
> >The analysis in Section 4.2 is poorly organized.
> >Line 504 claims that "the most advanced long-tailed learning methods no longer focus on improving a single strategy," but this claim is not well-supported by the preceding analysis.
>
> **A:** We sincerely apologize for the lack of clarity and structure in Section 4.2 of the initial submission. We have carefully revised this section to improve its organization, readability, and focus, ensuring that the discussions are logically connected and serve to provide actionable insights for designing better models. Below, we outline the revised structure and rationale behind the organization:
>
>
>
> ### **4.2.1 Overall Performance**
> - **Overall performance evaluation**: We begin with a discussion of the overall performance across methods and datasets to provide readers with a high-level understanding of the results.
> - **Curse of shot-based group evaluation**: We delve into the challenges posed by shot-based group evaluation, highlighting the trade-offs between head and tail class performance and their implications for model design and constraints for real-world applications (e.g., sensitivity to rare diseases).
>
> ### **4.2.2 Decoupled Components Discussion**
> - **Effectiveness of re-sampling across SOTAs**: We discuss and summarize how different re-sampling strategies affect state-of-the-art methods.
> - **Use two-stage training as a general paradigm**: Two-stage training is presented as an essential traning pipelines for the best performed methods in our benchmark, with observations on how it helps decouple backbone training from classifier adjustments. In this context, in the subsequent paragraphs, we focus on discussing solutions for improving feature representation training (e.g., MixUp and SSL) and enhancing classifier performance.
> - **MixUp can improve the feature representation**: We discuss how MixUp enhances feature learning and its consistent benefits across various models.
> - **Dilemmas of self-supervised learning for LTMIC**: We highlight the challenges and limitations of applying self-supervised learning (SSL) to long-tailed medical image classification (LTMIC).
> - **Modify classifier to reduce prediction bias**: We discuss the role of classifier modifications in mitigating prediction bias, particularly for tail classes, and connect this to the broader two-stage training paradigm.
>
> ### **4.2.3 Extended Tasks**
> - **LTMIC improves out-of-distribution detection**: We explore the potential of LTMIC approaches in out-of-distribution detection tasks and their practical significance in medical imaging, e.g., low-quality images, non-medical images and images with unknown rare diseases.
> - **Using imbalanced validation dataset for checkpoint selection**: We provide insights on why using an imbalanced validation set is more aligned with real-world scenarios and how it affects model selection.
> - **Multi-label classification is more challenging**: We discuss the unique challenges of multi-label classification, as detailed in the responses above.
>
> ###  **4.2.4 Overall Conclusion**
> - **What makes an essential strong baseline?**: We conclude our findings with actionable insights for building robust baselines in long-tailed medical image classification.
> - **Integration of medical domain prior knowledge**: We emphasize the importance of incorporating domain knowledge (e.g., hierarchical relationships) as a key driver for further improvements in medical imaging tasks.
>
> We hope that this revised structure and enhanced discussion will address the concerns.
>
> >Methods like ERM, cRT, and LWS represent, are not referenced properly.
> >Section 3.2.3 does not fully align with the table.
>
> **A:** **ERM** refers to `Empirical Risk Minimization`, which involves directly training on the original distribution using cross-entropy loss. **cRT** represents `classifier re-training`, and **LWS** refers to the `learnable weight-scaling classifier`; both methods are derived from *Kang et al., 2020.* [4] We will ensure the inclusion of the relevant references.
>
> For consistency between Table 2 and Sec. 3.2.3, we have clarified this in Line 314 - 315: ***We introduced and implemented over 30 methods but only present results for the most relevant ones to avoid redundancy and maintain clarity. Methods with similar or suboptimal performance were excluded to highlight those that best support our primary findings in line with the study's main motivation.*** The complete results are available in the supplementary documents.

---

> > ### Author Response · Authors · 2024-11-24
> > **Reply to Reviewer 3RRv (Part 6/N)**
> >
> > >The categorization of methods is confusing.
> >
> > **A:** Thank you for pointing this out. We would like to clarify the rationale behind our categorization of methods and address the potential confusion.
> >
> > As mentioned in the manuscript Line 226 - 228, ***Note that one methodology may contain more than one of those three taxonomy***. To provide a clear structure for the analysis, we grouped methods based on their primary motivation and technical contribution. This approach is consistent with some important reviews [5] in the field, which we referenced to ensure an objective and meaningful classification rather than relying on subjective judgment.
> >
> > We acknowledge the complexity introduced by overlapping methodologies and will ensure this point is more explicitly discussed in the revised manuscript to avoid any potential confusion.
> >
> > >The discussion on self-supervised learning (Line 398) appears out of place, as it is not introduced earlier.
> >
> > **A:** In the previous paragraph, we discussed the necessity of the **two-stage training** pipelines and emphasized the importance of enhancing the training of the **backbone/representation layer**, for example, through the use of MixUp. Following this, we explored the use of self-supervised learning (SSL) in long-tailed training within the natural image community and proposed that SSL could be a potential method to improve backbone representation training. This idea is intuitive because SSL does not rely on labeled data, and thus, without unsupervised signals.
> >
> > However, according to our experimental results, SSL performance was not satisfactory.
> >
> > **Table 3. Experimental results on ISIC2019-LT (r=100) trained from different self-supervised methods.**
> > | SSL Methods/Weights |     Source Data     | ISIC2019-LT (r=100) |
> > |:-------------------:|:-------------------:|:-------------------:|
> > | Supervised ImageNet |       ImageNet      |        59.33        |
> > |         BYOL        | ISIC2019-LT (r=100) |         16.70        |
> > |        MOCOv2       | ISIC2019-LT (r=100) |         25.60        |
> > |        SimCLR       | ISIC2019-LT (r=100) |         11.10        |
> > |         KCL         | ISIC2019-LT (r=100) |        50.83        |
> >
> > We provided several potential explanations for this outcome. Overall, the discussion of SSL serves two main purposes: first, we believe researchers in the community may find it valuable to explore the role of SSL methods in addressing long-tailed problems, which aligns with the two-stage training paradigm we advocate. Second, our codebase supports various backbone training strategies, and we consider SSL an essential component. This allows users to experiment with SSL for tuning hyperparameters, adjusting model structures, or refining training strategies.
> >
> > In the future work, we will explore some SSL methods designed for the **specific medical domain**, e.g., Li et al. for ophthalmology [6].
> >
> > >The mention of OOD detection (Line 421) is abrupt and lacks context.
> > >Could you elaborate on why the results of self-supervised learning and OOD detection are relevant in this paper?
> >
> > **A:** Similar to SSL, we also advocate that OOD detection is an extension task that will interest researchers working on long-tailed problems, especially considering the challenges that long-tailed issues bring to OOD detection tasks. Mathematically, OOD detection typically relies on the model's predictive confidence $p(y | x)$ to differentiate ID and OOD samples, but in long-tailed settings, this confidence distribution is skewed due to class imbalance.
> >
> > For head classes (frequent classes), the model is well-calibrated, producing high confidence:
> >
> > $
> > p(y_{head} | x_{head}) ≈ 1.
> > $
> >
> > For tail classes (rare classes), the model often produces low and uncertain confidence due to insufficient training data:
> >
> > $
> > p(y_{tail} | x_{tail}) ≪ 1.
> > $
> >
> > Consequently, the confidence for tail-class ID samples may overlap significantly with that of OOD samples:
> >
> > $
> > p(y_{tail} | x_{tail}) ≈ p(y_{OOD} | x_{OOD}),
> > $
> >
> > leading to poor separation between ID and OOD samples.
> >
> > In this context, methods designed for long-tailed problems adjust decision boundaries to account for class imbalances. This prevents the model from overly favoring head classes and ensures that tail classes are not confused with OOD samples. A well-calibrated model trained with long-tailed techniques is more likely to differentiate:
> >
> > $
> > p(y_{tail} | x_{tail}) > p(y_{OOD} | x_{OOD}).
> > $
> >
> > To conclude, Long-tailed problems in real-world scenarios aim to include as many categories as possible (e.g., rare diseases). However, there is still the potential for unknown categories to exist. The goal of OOD detection is to identify these unknown categories, thereby **achieving broader information coverage**. This aligns with the motivation of addressing long-tailed problems.

---

> ### Author Response · Authors · 2024-11-24
> **Reply to Reviewer 3RRv (Part 7/7, N=7)**
>
> > The section on using an imbalanced validation dataset for checkpoint selection is unclear about its purpose.
> > Figure 4 lacks labels for the x and y axes, making interpretation difficult.
>
> **A:** Thank you for highlighting these points. We would like to provide further clarification on the purpose of using an imbalanced validation dataset for checkpoint selection and the reasoning behind our conclusions.
>
> This approach addresses a key difference between long-tailed practices in natural images and medical images. In natural image datasets like CIFAR-LT and ImageNet-LT, the original data distribution is balanced, allowing researchers to construct balanced validation sets. This makes evaluation using metrics like loss value, macro accuracy, or group accuracy straightforward and fair. However, in medical imaging, creating a balanced validation set for checkpoint selection is extremely challenging due to the inherent imbalance in the data. Using an imbalanced validation set, while more challenging, better reflects real-world scenarios and aligns with the nature of long-tailed medical datasets.
>
> Additionally, sampling strategies in long-tailed tasks can lead to instability during training. For instance, some methods, e.g, RW, rely on fixed sampling weights determined at the beginning of training, while others adjust sampling dynamically during training, e.g, focal loss, causing fluctuations in performance. Therefore, beyond measuring a method's absolute performance, its stability during training is also a critical factor to consider.
>
> Given the lack of standardized evaluation methods in the community, this perspective ensures that checkpoint selection and performance evaluation are aligned with real-world data distributions, rather than relying solely on artificial or overly idealized validation settings.
>
> Regarding Figure 4, we apologize for the oversight. The labels for the `x` and `y` axes should be **AUROC value**. We will revise the figure. We also note that the conclusion regarding GCL exhibiting lower fluctuations highlights the importance of stability for practical deployment, which we will make more explicit in the revised manuscript.
>
> ## References
> [1] Ju, Lie, et al. "Hierarchical knowledge guided learning for real-world retinal disease recognition." IEEE Transactions on Medical Imaging (2023).
>
> [2] Wu, Tong, et al. "Distribution-balanced loss for multi-label classification in long-tailed datasets." ECCV 2020.
>
> [3] https://github.com/Westlake-AI/openmixup
>
> [4] Kang, Bingyi, et al. "Decoupling Representation and Classifier for Long-Tailed Recognition." International Conference on Learning Representations.
>
> [5] Zhang, Yifan, et al. "Deep long-tailed learning: A survey." IEEE Transactions on Pattern Analysis and Machine Intelligence 45.9 (2023): 10795-10816.
>
> [6] Li, Xiaomeng, et al. "Self-supervised feature learning via exploiting multi-modal data for retinal disease diagnosis." IEEE Transactions on Medical Imaging 39.12 (2020): 4023-4033.
>
> **We hope the above discussion will fully address your concerns about our work, and we would really appreciate it if you could be generous in raising your score.** We look forward to your insightful and constructive responses to further help us improve the quality of our work. Thank you!

---

> > ### Author Response · Authors · 2024-11-27
> > **Inquiry about further concerns/questions**
> >
> > Dear reviewer 3RRv,
> >
> > We would like to know if our response has addressed your concerns and questions. If you have any further concerns or suggestions for the paper or our rebuttal, please let us know. We would be happy to engage in further discussion and manuscript improvement.
> >
> > Thank you again for the time and effort you dedicated to reviewing this work.

---

> > ### Comment · Reviewer_3RRv · 2024-11-28
> >
> > Thank you for your detailed responses and for providing the repository. While this is a necessary step for any benchmark paper, it should be considered a baseline expectation rather than a distinguishing merit. Unfortunately, the repository raises new concerns.
> >
> > First, there are serious questions about the legality of data distribution. The repository includes a folder (MONICA-153F/numpy/) containing redistributed data files. The README provides dataset links but does not comply with the required licenses or provide proper credits. Even if the redistribution is legal, the repository should include a clear disclaimer.
> >
> > Second, there are numerous factual inaccuracies in the repository, which undermine its credibility. For example, CheXpert’s license is incorrectly listed as Apache License, WILDS-Camelyon17 is falsely linked to medmnist.com, and PathMNIST’s license is misidentified. These are fundamental errors that should have been verified before release. Benchmarks must uphold high standards of accuracy, as incorrect information can spread and lead to confusion and more misinformation. It is the responsibility of the scientific community to prevent the spread of such errors.
> >
> > Third, the code quality and design raise concerns about usability and extensibility. The codebase includes redundant components, such as standard models and metric calculations, complicating usage for researchers looking to extend it. The implementations lack proper references, making it unclear whether the code is original or adapted from other sources (it is difficult to imagine that none of the code was inspired by existing libraries). The coding style also reflects poor design choices; for instance, the 50-line if-else block in losses/get_loss.py could easily be replaced by a more concise approach. These issues contradict the claims of "flexibility" and "extensibility." I strongly recommend other reviewers examine the repository, as these problems are critical to the credibility of a benchmark.
> >
> > The primary concerns from my initial review remain unresolved. The motivation for the benchmark still fails to convince me. Addressing long-tail problems in medical imaging is important, but is a dedicated benchmark really needed for this? If so, does this benchmark's design genuinely reflect the challenges of medical long-tail problems? Do the methods compared meaningfully address these challenges? In my opinion, the answer to all these questions is "no." The authors need to clarify the value of this benchmark—whether it is intended to advance machine learning techniques or to address specific clinical issues. What meaningful questions can researchers answer by working on this benchmark, whether it is the authors' team or other researchers who follow?
> >
> > The experimental results, though extensive, are presented in a way that makes it difficult to derive meaningful conclusions. The tables lack proper references, and the discussion does not sufficiently highlight the strengths or weaknesses of the approaches. This leaves readers without actionable insights. The experiments feel more like a collection of numbers rather than meaningful insights. This aligns with my earlier critique about motivation: what can we learn from these experiments, and why are they important?
> >
> > While other reviewers have mentioned potential merits, such as public availability and thorough experiments, I believe that flawed public resources only add unnecessary confusion, and an overload of numbers without insights is akin to social media feeds—long on content but lacking substance.
> >
> > Overall, this benchmark appears hastily constructed, with significant issues in legality, accuracy, and implementation. While there is potential for improvement, the current version does not meet the standards for acceptance.

---

> ### Author Response · Authors · 2024-11-30
> **Reply to Reviewer 3RRV's Further Comments (Part 1/2)**
>
> We sincerely appreciate the reviewers' valuable time and effort, especially considering the detailed, point-by-point constructive feedback provided. We deeply respect these insights and have made every effort to address each comment in our response. We have not avoided any questions raised by the reviewers and have instead tried to answer them as comprehensively as possible. Once again, we thank the reviewers for their patience in reading our detailed response.
>
> > Legality of data distribution and factual inaccuracies in dataset details
>
> **A:** Thank you for pointing this out. We sincerely apologize for any misleading information regarding the data descriptions in the code repository.
>
> We will take this concern **very seriously**. Providing high-quality public resources is a responsibility we fully acknowledge, and we are committed to addressing the identified issues in legality, accuracy, and implementation.
> We will thoroughly audit and revise the dataset information in the repository, ensuring all licenses and links are accurate. For example:
>
> - Correcting CheXpert’s license details.
> - Properly linking WILDS-Camelyon17 to its original source.
> - Correctly identifying PathMNIST’s license.
>
>
> **Other Details**: Except for **CheXper**t, the distribution of all other datasets for non-commercial purposes is legal. CheXpert, however, prohibits distribution without written permission. It should be noted that the initial code repository we provided **did not include** any redistributed CheXpert data; we only provided processing scripts for reference. Additionally, all NumPy files **only contain sequences of filenames** and **did not modify any of the original data**, including images and labels.
>
> Additionally, we will include a clear **disclaimer** in the repository to guide users on license compliance and give proper credits for each dataset.
>
> > Code quality and design
>
> **A:** We appreciate the detailed feedback on the code quality and design. Our goal is to make the codebase as accessible and extensible as possible, and we take these suggestions seriously. We have revised most of the redundant components and will include proper citations to clarify whether specific implementations are original or adapted. Additionally, based on the suggestion from `@Reviewer FbkS`:
>
> `Avoid editorializing with value judgments: “benchmark is meticulously designed”; “we… develop a… well-structured codebase”; “our work provides valuable practical guidance”. Simply present your work and let the reader make these judgments!`,
>
> we have removed statements claiming contributions regarding the codebase to avoid causing any potential confusion.
>
> We also would like to emphasize that we do not see the publication of this work as the conclusion of the work on this code repository. Instead, we view it as a **starting point**. Our aim is to continuously maintain and improve the repository over time—for example, by incorporating new long-tailed learning methods, introducing emerging foundation models, supporting additional backbones, and exploring more diverse data augmentation techniques. While we acknowledge that the current code design has redundancies and limitations, it has been structured to allow for these future extensions. This is also reflected in the additional experimental results we provided during the rebuttal phase.

---

> > ### Author Response · Authors · 2024-11-30
> > **Reply to Reviewer 3RRV's Further Comments (Part 2/2)**
> >
> > >Motivation for the Benchmark
> >
> > **A:** Thank you for raising this critical point. We acknowledge that the motivation for the benchmark could be more clearly articulated.
> >
> > While we have conducted a unified analysis of all methods across different datasets and provided quantitative conclusions on methodological performance, we also presented some findings that **differ from the analysis of long-tailed problems in natural images**:
> >
> > ### On Data
> > - **Data distribution:** In long-tailed learning studies for natural images, most datasets are resampled according to certain distributions, such as Pareto distribution. In our study, we also retained the real-world data distributions in medical datasets, such as KVASIR, ODIR, RFMiD, CheXpert, and WILDS. For other datasets, even when our datasets are resampled according to the Pareto distribution, the order of disease frequencies still reflects the real-world scenario.
> >
> > - **Resampling strategies:** Head classes in medical datasets exhibit greater variance, while rare diseases are often highly similar [1]. This means that performance does not always improve with more samples, presenting challenges for designing sampling strategies. We highlighted and discussed these challenges by analyzing the sampling strategies for class imbalance (e.g., CBLoss) and learning difficulty of samples (e.g., Focal loss).
> >
> > ### On Methodologies
> > - **Pretraining is crucial for medical images:** While methods for natural images often train from scratch, we emphasised the importance of pre-training for medical image tasks, whether using ImageNet or domain-specific foundation models.
> >
> > - **SOTA performance variability:** Intuitively, in natural image tasks, newer methods often demonstrate better performance. However, for medical images, the uniformity of medical imaging (e.g., diagnostics often depend on foreground features rather than the majority of background similarity) and the fine-grained differences in certain disease features (e.g., lesion variation leading to different diagnoses) result in some methods being less effective, for instance, which are typically designed to handle larger-scale features or pattern variations, but may struggle to effectively capture these subtle yet critical fine-grained differences.
> > - **Incorporating prior medical knowledge** We have placed significant emphasis on the importance of incorporating domain-specific medical knowledge, e.g, hierarchical or other semantic relationships among diseases.This additional layer of complexity requires adapted methodologies that leverage domain-specific insights, further distinguishing medical imaging from other domains.
> >
> > - **Self-Supervised Learning (SSL):** We proposed SSL as an important strategy for addressing long-tailed problems. However, through specific experiments, we also demonstrated its limitations on medical datasets and provided a discussion to contextualize these findings.
> >
> > ### On Evaluation
> >
> > **Challenges in shot-based group evaluation:** We discussed potential issues with using shot-based group metrics in medical scenarios. For example, while GCL achieves the best performance in most scenarios, it does so at the expense of head-class performance. This highlights a potential flaw in the current evaluation design. In medical scenarios, such trade-offs are often necessary rather than aiming for purely optimal metrics. To our knowledge, this perspective has been largely overlooked in the existing literature on long-tailed learning for medical imaging. We discussed the challenge of using an imbalanced validation set and its implications in our paper.
> >
> >
> > **Extended tasks:** Beyond self-supervised learning and the long-tailed problem itself, we also explored additional tasks relevant to medical applications, such as OOD detection. We acknowledge that our earlier version provided an insufficient description of this task, but we have supplemented the content in our rebuttal and will include it in the revised manuscript.
> >
> > Finally, we would like to acknowledge that **our expertise cannot fully encompass all medical domains**, making it difficult to provide finer-grained discussions for certain datasets or medical domains. However, we have sought to introduce additional methodological insights to explore optimization opportunities. For example, we supplemented our response with discussions on using different data augmentations across various datasets to highlight potential areas for improvement.
> >
> > We sincerely thank you once again for the constructive feedback, which has helped us refine and improve our work significantly. We are committed to addressing these issues in the revised manuscript and ensuring that our contributions provide value to the community.
> >
> > We kindly hope the reviewer might reconsider their score in light of the substantial revisions and improvements we have made to address the raised concerns.
> >
> > [1] Flexible Sampling for Long-tailed Skin Lesion Classification.

---

### Official Review · Reviewer_nS7a · 2024-11-04

**Soundness:** 2
**Presentation:** 2
**Contribution:** 3
**Rating:** 5
**Confidence:** 4

**Summary:**

The authors introduce the problem of long-tailed medical image classification and challenges in the field. Then they develop MONICA which is a package to benchmark various methods ranging from different loss functions, augmentations, etc on the benchmark datasets across medical image classification tasks. They provide an overview of datasets and methods and experiment results on the datasets. The authors additionally share learnings and observations.

**Strengths:**

- A good overview of the datasets curated for this work
- important contribution of decoupling the codebase
- A good overview of the method approaches
- practically useful to AI researchers in medical imaging

**Weaknesses:**

- It would help to expand the benchmark datasets and bring in a canonical set for a field such as Camlyon for Pathology, etc. WILDS (medical subset) is a great example of a dataset to bring in to this benchmarking codebase
- Resnet-50 is used as a backbone but the community has generally moved on to more complex backbones such as ConvNext / Swin or foundation model backbones for different datasets.
- Generally the community uses pretrained backbones rather than training the backbones from the scratch.
- The same backbone is used for every task for fairness but generally a sweep over backbones would help since different modalities and tasks require different approaches
- Top-1 accuracy is an in appropriate metric for model selection in imbalances settings and AUROC, AUPRC, F1 should be used
- error bars are missing in experiments
- More thorough error analysis
- Clearer articulation of novel insights
- Better connection to clinical relevance
- More detailed ablation studies

**Questions:**

- Are their any key trends that you'll observed across the board to narrow down the design space for the future across the general task space? The results are not convincing in any one direction across the board on tasks and methods
- Do you'll think stronger backbones can help learn better features?
- Did you'll consider trying complex augmentation techniques such as AugMix or even learned augmentations?

---

> ### Author Response · Authors · 2024-11-23
> **Reply to Reviewer nS7a (Part 1/N)**
>
> We thank the reviewer for the encouraging feedback and helpful suggestions.
>
> We have conducted extensive updates and additional experiments based on the reviewers' valuable feedback. Specifically,
>
> - We added an anonymous repository for our codebase: https://anonymous.4open.science/r/MONICA-153F/README.md
> - We introduced new datasets and transformed existing ones (e.g., WILDS) to better align with long-tailed settings (in progress).
> - We added support for more advanced backbones, including ViT, Swin Transformer, ConvNext, and RetFound, and performed detailed experiments to evaluate their impact.
> - We expanded the evaluation metrics to include AUROC, AUPRC, and F1 Score, providing a more comprehensive assessment of model performance.
> - Additional ablation studies were conducted to analyze input size variations, backbone comparisons, and augmentation techniques such as RandAug and AugMix.
>
> The detailed responses are outlined below:
>
> ### Weakness
>
> > More datasets Pathology such as WILDS (medical subset).
>
> **A:** Thank you for your valuable insights on pathology datasets. We have chosen PathMNIST as a representative dataset for this study. Our codebase supports customized datasets, enabling researchers to conduct their own experiments.
>
> Regarding WILDS, we observed that it processes the **original whole-slide images** into **patches** to identify the presence of tumors, resulting in a **binary classification** task. However, this setup may not align well with our long-tailed setting. To address this, we used the **"stage label"** as the new patch annotation, transforming the task into a **9-class** classification problem.
>
> The experiments are currently in progress as we prioritized supplementing some other important experimental results within the limited time. The complete results will be updated in the final revision. Thank you for your patience and understanding.
>
> > More complex backbones such as ConvNext / Swin or foundation model backbones for different datasets.
>
> **A:** Unless explicitly specified in the original paper that a particular backbone must be used (e.g., **RSG uses ResNext**), we choose ResNet as our backbone for the following reasons: the methods supported by our benchmark predominantly use ResNet as the backbone in their original papers. This is especially important for methods that require modifications to the network structure (e.g., the **ResNet** in **SADE**). Attempting to modify ViT in the same way as ResNet to achieve a similar motivation could introduce discrepancies, potentially leading to **unfair evaluations**.
>
> However, we appreciate the suggestions from the reviewer to add more backbones with different architectures. Our updated codebase now supports more advanced backbones such as ViT, Swin Transformer, ConvNext, as well as relevant foundation models like RetFound for ophthalmology. We have provided corresponding alerts in the codebase for those fixed backbones proposed by the original paper. We completed training for **ViT, Swin Transformer, and ConvNext** on the ISIC dataset. Here, we present partial results, and the complete results will be included in the appendix.
>
> **Table 1. ISIC2019-LT (r=100)**
> |     Methods     | ResNet50 | ViT-Base | SwinTrans-Base | ConvNext-Base |
> |:---------------:|:--------:|:--------:|:--------------:|:-------------:|
> |       ERM       |  59.33   |  53.17   |     68.27      |     74.05     |
> |        RS       |  60.06   |  59.55   |     72.41      |     77.14     |
> |        RW       |  59.00   |  38.09   |     31.27      |     63.32     |
> |      Focal      |  57.28   |  35.86   |     71.05      |     63.90     |
> |     CB-Focal    |  60.44   |  40.93   |     52.84      |     52.43     |
> |     LADELoss    |  58.17   |  53.17   |     73.71      |     75.80     |
> |       LDAM      |  58.61   |  32.79   |     51.43      |     70.68     |
> | BalancedSoftmax |  59.28   |  56.51   |     60.98      |     69.32     |
> |      VSLoss     |  56.67   |  56.37   |     73.71      |     56.18     |
> |      MixUp      |  54.78   |  27.95   |     73.12      |     63.65     |
> |      MiSLAS     |  55.83   |  54.43   |     64.98      |     67.82     |
> |       GCL       |  64.06   |  57.27   |     70.07      |     72.73     |
> |       cRT       |  63.28   |  54.41   |     59.00      |     76.10     |
> |       LWS       |  56.83   |  59.84   |     67.98      |     77.37     |
> |       KNN       |  61.44   |  56.78   |     65.06      |     66.39     |
> |       LAP       |  61.22   |  44.85   |     73.38      |     73.95     |
> |   De-Confound   |  59.67   |  55.62   |     72.18      |     57.91     |
> |     DisAlign    |  64.44   |  56.17   |     72.13      |     80.64     |

---

> > ### Author Response · Authors · 2024-11-23
> > **Reply to Reviewer nS7a (Part 2/N)**
> >
> > **Table 2. ISIC2019-LT (r=200)**
> > |     Methods     | ResNet50 | ViT-Base | SwinTrans-Base | ConvNext-Base |
> > |:---------------:|:--------:|:--------:|:--------------:|:-------------:|
> > |       ERM       |  54.06   |  28.70   |     55.28      |     65.02     |
> > |        RS       |  58.33   |  50.91   |     52.97      |     71.92     |
> > |        RW       |  56.50   |  32.74   |     19.02      |     43.21     |
> > |      Focal      |  54.50   |  29.28   |     63.97      |     56.86     |
> > |     CB-Focal    |  58.50   |  46.28   |     53.70      |     58.29     |
> > |     LADELoss    |  51.67   |  48.90   |     66.86      |     63.28     |
> > |       LDAM      |  54.94   |  55.25   |     49.61      |     55.06     |
> > | BalancedSoftmax |  61.89   |  49.85   |     65.24      |     67.89     |
> > |      VSLoss     |  53.39   |  28.70   |     66.86      |     68.52     |
> > |      MixUp      |  50.33   |  51.97   |     67.39      |     70.89     |
> > |      MiSLAS     |  56.50   |  56.92   |     51.00      |     70.40     |
> > |       GCL       |  64.00   |  53.81   |     68.09      |     68.99     |
> > |       cRT       |  60.22   |  60.55   |     72.20      |     67.67     |
> > |       LWS       |  54.50   |  51.83   |     57.11      |     68.39     |
> > |       KNN       |  58.33   |  56.28   |     62.72      |     61.50     |
> > |       LAP       |  58.39   |  54.98   |     66.86      |     63.26     |
> > |   De-Confound   |  53.17   |  55.25   |     64.98      |     57.36     |
> > |     DisAlign    |  63.33   |  48.15   |     64.80      |     57.48     |
> >
> > **Table 3. ISIC2019-LT (r=500)**
> > |     Methods     | ResNet50 | ViT-Base | SwinTrans-Base | ConvNext-Base |
> > |:---------------:|:--------:|:--------:|:--------------:|:-------------:|
> > |       ERM       |  45.78   |  47.16   |     48.98      |     61.35     |
> > |        RS       |  47.39   |  42.36   |     63.29      |     60.05     |
> > |        RW       |  44.78   |  27.48   |     16.89      |     47.84     |
> > |      Focal      |  46.67   |  26.06   |     56.27      |     63.34     |
> > |     CB-Focal    |  48.39   |  35.27   |     17.33      |     42.82     |
> > |     LADELoss    |  45.17   |  47.16   |     41.12      |     56.33     |
> > |       LDAM      |  45.72   |  50.15   |     50.41      |     42.18     |
> > | BalancedSoftmax |  51.11   |  35.35   |     49.32      |     55.70     |
> > |      VSLoss     |  45.89   |  47.16   |     41.12      |     33.87     |
> > |      MixUp      |  44.33   |  45.32   |     30.44      |     40.46     |
> > |      MiSLAS     |  48.61   |  50.54   |     30.62      |     61.52     |
> > |       GCL       |  54.83   |  50.75   |     49.02      |     52.68     |
> > |       cRT       |  46.61   |  48.27   |     50.31      |     60.79     |
> > |       LWS       |  47.00   |  49.50   |     47.67      |     59.38     |
> > |       KNN       |  49.22   |  44.44   |     55.72      |     56.06     |
> > |       LAP       |  49.56   |  42.53   |     57.02      |     59.72     |
> > |   De-Confound   |  42.50   |  48.72   |     53.26      |     61.61     |
> > |     DisAlign    |  51.72   |  44.37   |     46.39      |     49.01     |
> >
> > **The Key Observations:**
> >
> > - **ViT-Base struggles** in most settings and methodologies compared to ConvNext and SwinTrans. This is due to that transformer relies on **more datasets** for training.
> > - But the transformer structure is not necessarily worse than CNN. Swin Transformer base achieves higher performance than the ResNet50 in terms of most settings and methods.
> > - **ConvNext-Base** consistently outperforms other backbones (ResNet50, ViT-Base, and SwinTrans-Base) across many methods, especially for more complex scenarios. This suggests that modern architectures like ConvNext-Base are more robust for handling diverse long-tailed datasets.
> > - Methods involving **model modifications** often experience **significant performance degradation** or **show less noticeable improvements** when applied to **new backbones** (i.e., backbones different from those used in the original papers). For example, when adapting `GCL` (which performs well on most datasets using `ResNet50`), modifying `ResNet50` typically involves directly adjusting the `model.fc` layer. However, when adapting `GCL` to `Swin Transformer`, there are two possible approaches: either modify the entire `classifier head` or adjust only the `fc` layer within the `classifier head`. Here, we present the results of the better-performing approach for each case, but no unified conclusion supports whether the former or the latter is superior. Therefore, for methods that involve network modifications, it is still recommended to **use the network architecture specified in the original paper**.

---

> ### Author Response · Authors · 2024-11-23
> **Reply to Reviewer nS7a (Part 3/N)**
>
> >Generally the community uses pretrained backbones rather than training from the scratch.
>
> **A:** Unless otherwise specified (e.g., for **SADE** or other **foundation models**), our methods use backbones pre-trained on ImageNet. In contrast, the community  often employs training from scratch, as datasets like **CIFAR-10/100** and **ImageNet-LT** are widely regarded as important benchmark datasets.
>
> >The same backbone is used for every task for fairness but generally a sweep over backbones would help since different modalities and tasks require different approaches.
>
> **A:** As our response to **More Backbones**, our updated codebase supports additional backbones, and we will include discussions based on the results. Furthermore, we have added a set of experiments with varying **input sizes**. Partial results are shown below, and we believe these experiments provide valuable insights into how different input sizes impact performance across various datasets and methods.
>
> >Top-1 accuracy is an in appropriate metric for model selection in imbalances settings and AUROC, AUPRC, F1 should be used.
>
> **A:** In long-tailed learning, the community typically uses **shot-based group Top-1 Accuracy** as a key metric to evaluate models. We hope the reviewers understand that this choice is based on the findings of **supported methods from relevant references** and **some important surveys**, e.g, *Zhang et al.* [1], rather than being a subjective decision. However, we believe adding more metrics can help users better understand the differences between methods in practical applications. For example, in clinical scenarios, many use cases may prioritize **sensitivity**, whereas screening scenarios often focus on a balance between **sensitivity** and **specificity**. To address this, our updated codebase includes additional metrics such as **AUROC, AUPRC, and F1 Score**.
>
> >Error bars are missing in experiments.
>
> **A**: Apologies for the missing details. We ran each methodology on each dataset **three times** using different random seeds. In the updated version, we will include error bars (e.g., rounded to one decimal place) for greater clarity. Additionally, we plan to establish a website to provide more comprehensive results, offering a better overall understanding of this benchmark.
>
> >More thorough error analysis, clearer articulation of novel insights, better connection to clinical relevance, more detailed ablation studies.
>
> **A:** We sincerely appreciate the reviewers’ constructive feedback regarding the need for more thorough analysis and discussion.
>
> We will include a deeper investigation into error cases, particularly focusing on challenging classes in long-tailed datasets, to provide better insights into model limitations and potential improvements. We will contextualize our results with respect to real-world medical applications, discussing how our findings could be utilized in practice and their potential impact on clinical workflows (e.g., with experiments on augmentations).
>
> These suggestions are invaluable for improving the quality and impact of our work.
>
> ### Questions:
>
> >Key trends observed across the board?
>
> **A:** As discussed in the experimental section, we conducted an in-depth analysis from multiple perspectives, including the use and summarization of **re-sampling strategies**, employing appropriate data augmentation (e.g., MixUp) to enhance the **backbone's representation capability**, improving **classifiers** to mitigate distribution bias, and the necessity of **two-stage training methods**. Through these analyses, we not only derived many significant conclusions but also identified **potential improvement directions** for each component, providing a broader design space for future researchers to develop new methods.
>
> Furthermore, we would like to emphasize that the codebase we provide enables the design, improvements and potential combination of modules, offering greater flexibility. The additional experiments we conducted with **different backbones and data augmentation strategies** also release new insights.
>
> Additionally, we explored the extension tasks on key issues relevant to this field, such as **OOD detection**. At the same time, we highlighted the importance of the intrinsic characteristics of medical data, such as incorporating prior knowledge as auxiliary information—for example, hierarchical information[2], which serves as an excellent case in point.
>
> >Do stronger backbones can help learn better features?
>
> **A:** Based on the experimental results with the additional backbones, it is evident that stronger backbones can indeed help learn better features. However, due to differences in network architecture (**CNN-based, transformer-based**) and the number of parameters, it is challenging to directly compare their differences horizontally. To address this, we added a new set of results to specifically compare the performance differences between **ResNet50** and **ResNet101** across methods.

---

> ### Author Response · Authors · 2024-11-23
> **Reply to Reviewer nS7a (Part 4/N)**
>
> **Table 4. ISIC2019-LT (r=100)**
> |     Methods     | ResNet50 | ResNet101 |
> |:---------------:|:--------:|-----------|
> |       ERM       |  59.33   | 59.25     |
> |        RS       |  60.06   | 63.94     |
> |        RW       |  59.00   | 55.18     |
> |      Focal      |  57.28   | 60.13     |
> |     CB-Focal    |  60.44   | 59.14     |
> |     LADELoss    |  58.17   | 62.59     |
> |       LDAM      |  58.61   | 59.93     |
> | BalancedSoftmax |  59.28   | 63.08     |
> |      VSLoss     |  56.67   | 44.51     |
> |      MixUp      |  54.78   | 65.35     |
> |      MiSLAS     |  55.83   | 59.06     |
> |       GCL       |  64.06   | 72.22     |
> |       cRT       |  63.28   | 56.60     |
> |       LWS       |  56.83   | 36.69     |
> |       KNN       |  61.44   | 63.19     |
> |       LAP       |  61.22   | 60.88     |
> |   De-Confound   |  59.67   | 60.21     |
> |     DisAlign    |  64.44   | 65.27     |
>
> **Table 5. ISIC2019-LT (r=200)**
> |     Methods     | ResNet50 | ResNet101 |
> |:---------------:|:--------:|-----------|
> |       ERM       |  54.06   | 56.16     |
> |        RS       |  58.33   | 58.58     |
> |        RW       |  56.50   | 47.16     |
> |      Focal      |  54.50   | 50.29     |
> |     CB-Focal    |  58.50   | 53.37     |
> |     LADELoss    |  51.67   | 39.30     |
> |       LDAM      |  54.94   | 55.95     |
> | BalancedSoftmax |  61.89   | 58.76     |
> |      VSLoss     |  53.39   | 43.53     |
> |      MixUp      |  50.33   | 61.46     |
> |      MiSLAS     |  56.50   | 54.42     |
> |       GCL       |  64.00   | 63.86     |
> |       cRT       |  60.22   | 59.96     |
> |       LWS       |  54.50   | 44.55     |
> |       KNN       |  58.33   | 59.11     |
> |       LAP       |  58.39   | 60.45     |
> |   De-Confound   |  53.17   | 55.88     |
> |     DisAlign    |  63.33   | 60.26     |
>
> **Table 6. ISIC2019-LT (r=500)**
> |     Methods     | ResNet50 | ResNet101 |
> |:---------------:|:--------:|-----------|
> |       ERM       |  45.78   |   52.71   |
> |        RS       |  47.39   |   45.90   |
> |        RW       |  44.78   |   37.24   |
> |      Focal      |  46.67   |   33.18   |
> |     CB-Focal    |  48.39   |   37.47   |
> |     LADELoss    |  45.17   |   23.72   |
> |       LDAM      |  45.72   |   49.82   |
> | BalancedSoftmax |  51.11   |   50.54   |
> |      VSLoss     |  45.89   |   32.99   |
> |      MixUp      |  44.33   |   57.21   |
> |      MiSLAS     |  48.61   |   47.36   |
> |       GCL       |  54.83   |   59.86   |
> |       cRT       |  46.61   |   55.52   |
> |       LWS       |  47.00   |   38.93   |
> |       KNN       |  49.22   |   49.11   |
> |       LAP       |  49.56   |   51.16   |
> |   De-Confound   |  42.50   |   50.29   |
> |     DisAlign    |  51.72   |   42.99   |
>
> The key observations:
> - Overall, increasing the number of parameters does improve performance for most methods, particularly for MixUp. This is likely because MixUp, as a data augmentation and regularization technique, benefits from greater model capacity, allowing it to learn more robust feature representations.
>
> - In terms of efficiency, as the level of imbalance increases, simply increasing model capacity shows diminishing returns, indicating the need to explore other strategies to enhance performance. Additionally, under the `Pareto distribution sampling setting`, the increase in imbalance significantly reduces the number of training samples, which may lead to **overfitting** in larger models. Therefore, **the number of model parameters** should be appropriately aligned with **the amount of training data** to achieve optimal performance.
>
> >Add augmentation techniques such as AugMix?
>
> **A:** The current version of codebase supports RandAug and we plan to support some other augmentation codebase such as **OpenMixup**[3] in the future update.
>
> We have also tested different augmentation techniques:
>
> **Table 7. Results with different augmentation techniques**
> |                   | Weak (Resize) | Crop (Strong) | RandAug | AugMix + Crop |
> |:-----------------:|:-------------:|:-------------:|:-------:|:-------------:|
> | ISIC-2019 (r=100) |     58.72     |     64.06     |  65.89  |     66.34     |
> |       KVASIR      |     86.07     |     85.14     |  87.88  |     85.32     |
> |    OrganAMNIST    |     79.09     |     78.21     |  80.13  |     79.06     |

---

> ### Author Response · Authors · 2024-11-23
> **Reply to Reviewer nS7a (Part 5/5, N=5)**
>
> **The key observations**:
>
> - Using more **complex data augmentation** methods like RandAug and AugMix can further improve performance. For datasets with diverse image patterns, such as dermoscopy images (e.g., with color variations), the performance gains from applying complex augmentations are often greater compared to others such as radiology datasets.
> - Cropping is a common method in data augmentation, but it may not be suitable for all medical datasets, as medical data often relies on **specific regions of lesions** for classification or diagnosis, such as in cases of diabetic retinopathy grading.
> - Skin lesions are often characterized by certain specific local features, such as the shape of the lesion, irregular edges, and color distribution. By applying **cropping**, the model can focus more on the detailed features of the lesion area, reducing background interference and thereby improving classification or detection performance.
>
> ### References
> [1] Deep Long-Tailed Learning: A Survey.
>
> [2] Hierarchical knowledge guided learning for real-world retinal disease recognition.
>
> [3] https://github.com/Westlake-AI/openmixup
>
> **We hope the above discussion will fully address your concerns about our work, and we would really appreciate it if you could be generous in raising your score.** We look forward to your insightful and constructive responses to further help us improve the quality of our work. Thank you!

---

> > ### Comment · Reviewer_nS7a · 2024-11-26
> >
> > Thank you so much for your detailed responses! I will raise my score to reflect improved confidence

---

> > > ### Author Response · Authors · 2024-11-27
> > > **Thank you for increasing your score!**
> > >
> > > Thank you for upgrading your score! We appreciate the time and effort you dedicated to reviewing this work.

---

### Author Response · Authors · 2024-12-02
**Official Comment by Authors**

**Dear Reviewers,**

We would first like to express our gratitude for your positive comments and support for this work:

- The datasets, methodologies, and figures are well-formulated, clearly presented, and systematically organized (`Reviewer nS7a and FbkS`).
- This work establishes a standardized and reproducible benchmark for long-tailed medical image classification, addressing evaluation variability and promoting fair comparisons to advance research in this field (`All Reviewers`).
- The contribution to the codebase is notable. The framework is publicly available and designed to serve as an extensible resource for future research in long-tailed medical image classification (`Reviewer nS7a, gxGu, and FbkS`).
- The experiments are comprehensive, covering a wide range of relevant methods, datasets, and tasks (`Reviewer gxGu and FbkS`).
- The discussion is insightful, going beyond merely presenting benchmark results (`Reviewer FbkS`).

We also sincerely appreciate your constructive feedback and valuable suggestions, which have significantly contributed to improving our work. We believe that our responses during the discussion phase have sufficiently addressed most of the concerns raised. Below, we outline the key revisions and additions we have implemented:

- **Clarification on the motivation** for building a long-tailed medical image learning benchmark (`Reviewer 3RRv`).
- **Provided an anonymous repository** for our codebase: [https://anonymous.4open.science/r/MONICA-153F/README.md](https://anonymous.4open.science/r/MONICA-153F/README.md) (`Reviewer nS7a and 3RRv`).
- **Added support for advanced backbones and foundation models,** including ViT, Swin Transformer, ConvNext, and RetFound, accompanied by detailed experiments to evaluate their impact (`Reviewer nS7a and gxGu`).
- **Added support for more evaluation metrics,** including AUROC, AUPRC, and F1 Score (`Reviewer nS7a and FbkS`).
- **Supplemented domain-specific analysis and results,** including an investigation into the effect of different augmentation techniques on various datasets, accompanied by discussions incorporating medical insights (`Reviewer nS7a and 3RRv`).
- **Supplemented results on self-supervised learning (SSL)** to provide further analysis and address its limitations (`Reviewer 3RRv and FbkS`).
- **Enhanced the motivation and definition** of relevant extended tasks, such as out-of-distribution (OOD) detection (`Reviewer 3RRv and FbkS`).
- **Expanded discussions on multi-label long-tailed learning** to explore its unique challenges and differences from multi-class settings (`Reviewer 3RRv`).


Thank you for your time and thoughtful consideration.

**Sincerely,**

*The Authors*

---

### Meta-Review · Area_Chair_pDE7 · 2024-12-23

**Metareview:**

The paper introduces MONICA (Medical OpeN-source Long-taIled ClassifiCAtion), a comprehensive benchmark designed for long-tailed medical image classification (LTMIC). MONICA integrates 12 publicly available datasets across six medical domains and implements over 30 long-tailed (LT) learning methods. The framework provides a unified, reproducible platform for evaluating LT learning strategies, aiming to standardize comparisons and advance research in this challenging problem. The authors present a detailed overview of datasets, methods, and experimental results, along with observations and learnings.

Strength: The benchmark addresses the variability in dataset and evaluation settings, offering a fair and standardized framework for LT learning in medical imaging. It covers diverse datasets and strategies, making it practically useful for researchers. The publicly available, extensible codebase encourages further advancements in LTMIC research.

Weakness: The paper does not introduce new datasets or methods, relying on existing publicly available datasets, many of which are derived from prior work (e.g., MedMNIST). Experimental analysis lacks domain-specific insights:the use of a single backbone across tasks, while ensuring fairness, limits exploration of modality-specific optimizations. The results are presented in a way which may be hard to derive clear conclusions from.

Overall, this work could be potentially useful and helpful to standardize comparisons and advance research in long-tailed learning in the medical image research community. Though during the rebuttal, the manuscript has been improved a lot based on the reviewers’ comments including adding more advanced model backbones, more comprehensive evaluation metrics, more compared baselines, I agree with the reviewer’s comment that a more thorough major revision would be appreciated to further refine the work. Thus, I recommend reject based on the current version.

**Additional Comments On Reviewer Discussion:**

During the rebuttal, the author provided a very detailed and extensive response to all reviewers’ comments, which are highly appreciated. Most reviewers have responded to the rebuttal and some reviewers have raised the score to reflect the paper improvement from the author’s response and manuscript revision, though some concerns still remain.

---

### Decision · Program_Chairs · 2025-01-22

Reject